# Hydrodynamics of a high Alpine catchment characterized by four natural tracers

Anthony Michelon[1], Natalie Ceperley[2], Harsh Beria[3], Joshua Larsen[4,5], Torsten Vennemann[1], Bettina Schaefli[2]

[1]Institute of Earth Surface Dynamics (IDYST), Faculty of Geosciences and Environment (FGSE), University of Lausanne, Lausanne, Switzerland

[2]Institute of Geography (GIUB) and Oeschger Center of Climate Change Research (OCCR), University of Bern, Bern, Switzerland.

[3]Department of Environmental Systems Science, ETH Zurich, Zurich, Switzerland

[4]School of Geography, Earth and Environmental Sciences, University of Birmingham, Birmingham, UK

[5]Birmingham Institute for Forest Research (BIFOR), University of Birmingham, Birmingham, UK

*Correspondence to*: Natalie Ceperley (natalie.ceperley@giub.unibe.ch)

## Abstract

Hydrological processes in high-elevation catchments are strongly influenced by alternating snow accumulation and melt in addition to summer rainfall. Although diverse water sources and flow paths generate streamflow in the world's "water towers" emerge from these two driving inputs, a detailed process understanding remains poor. We measured a combination of natural tracers of water at a high frequency, including stable isotope compositions, electrical conductivity (EC), and water and soil temperature to characterize hydrological processes in a snow-dominated Alpine catchment and to understand the diversity of streamflow sources and flow paths. Stable isotope composition of the sampled water revealed the prominence of snowmelt year-round (even during winter baseflow) and a strong flushing of the entire system with snowmelt at the start of the main melt period, sometimes referred to as the freshet, led to a reset, or return to baseline, of the isotopic values in most sampled water. Soil temperature measurements help identify snow-free periods and indicate sub-snowpack local flow, for example in the case of rain-on-snow events. Water temperature measurements in springs can indicate flow path depth. EC measurements reflect the magnitude of subsurface exchange and allow for the separation of subsurface snowmelt contribution to streamflow from the contribution of stored groundwater. These insights into the details of streamflow generation in such a dynamic environment were only made possible due to the intense, year-round water sampling. The sampled tracers are revealed to complement each other in important ways particularly because they were sampled during winter and spring, both snow-covered periods, the importance of which is a key implication of this work.

# 1    Introduction

Hydrology in Alpine environments is largely dominated by snow accumulation and melt processes with ensuing high sensitivity to changes in climate (Hanus et al., 2021). For Alpine catchments with a mean elevation above approximately 1500 masl (Santos et al., 2018), winter snowfall leads to the build-up of a seasonal snowpack, which results in low flow occurring between November and March (Schaefli et al., 2013) and maximum monthly streamflow related to melt between May and August, depending on the depth and extent of the seasonal snowpack and on the degree of glacier cover (Hanus et al., 2021; Muelchi et al., 2021). Given the importance of these cycles of accumulation and melt and the resulting streamflow regime for water resource availability, an important body of literature focuses on quantifying the streamflow regime in such environments, either based on streamflow observations (Blahusiakova et al., 2020; Brunner et al., 2019; Musselman et al., 2021; Hammond and Kampf, 2020) or modelling (Foster et al., 2016; Livneh and Badger, 2020; Muelchi et al., 2021).

Detailed hydrological process studies in high Alpine catchments remain, however, relatively rare even if detailed insights into the fate of rainfall and snowmelt in such catchments are required for model-based extrapolations of their hydrological response into the future, given the likely changes in climate. In addition to logistical challenges, continuous monitoring in frozen environments require development of specific methods and equipment (Rucker et al., 2019), which explains the small number of studies in such places.

Existing field-based studies can be classified according to their focus: i) understanding dominant runoff generation mechanisms during rainfall and snowmelt events (Penna et al., 2016; Engel et al., 2016), including small-scale studies of snowpack flow paths (Webb et al., 2020); ii) understanding the origin of winter low flow (Floriancic et al., 2018); iii) quantification of groundwater or spring recharge (Lucianetti et al., 2020) and seasonal groundwater storage (Arnoux et al., 2020); or iv) understanding the role of glaciers and rock-glaciers in the hydrological response of high elevation catchments (Brighenti et al., 2019; Zuecco et al., 2019; Ohlanders et al., 2013; Penna et al., 2014). A common feature of these studies is the use of natural tracers, such as electric conductivity or stable isotope compositions of water, to gain new insights into the fate of rainfall and snowfall and related water flow paths and to formulate hypotheses about dominant runoff drivers at specific times of the year, or the hydrologic response of selected landscape units.

To help fill some of the important knowledge gaps on elevational and seasonal drivers, this work uses high frequency tracer sampling to quantify drivers of the hydrologic response of a high elevation catchment throughout all streamflow periods, ranging from winter low flow to different stages of the melt season and the autumn recession by compiling and complementing observational data from the intensively studied Vallon de Nant catchment in the Swiss Alps (Benoit et al., 2018; Giaccone et al., 2019; Ceperley et al., 2020; Mächler et al., 2021; Michelon et al., 2021b; Thornton et al., 2021; Beria et al., 2020; Antoniazza et al., 2022).

The overall objective of this work is to examine dominant hydrological processes and associated flow paths in Vallon de Nant during different periods of the year through the lenses of four tracers: soil temperature, water temperature, electrical conductivity, and stable isotope composition of water. The first tracer used in this study is the stable isotope composition of

water, a natural tracer that has been extensively used to characterize snow hydrological processes (e.g. Beria et al., 2018), and is particularly useful to examine the interplay between different water compartments (rainfall, snowpack, springs, groundwater), recharge and evaporation processes (e.g. Sprenger et al., 2016). Electrical conductivity measurements as an additional tracer provides information on subsurface flow paths and water residence times in the subsurface (Cano-Paoli et al., 2019). Water temperature measurements can be used to quantify connectivity between water sources and the atmosphere (Constantz, 2008). And lastly, soil temperature measurements are used to identify periods of thermal insulation from the seasonal snow cover (Trask et al., 2020).

Using these tracers, we explore the origin of winter streamflow (from groundwater or from localized snow melt) (Floriancic et al., 2018; Hayashi, 2020), the dominant runoff processes that drive streamflow generation during early compared to late snow melt phase (Brauchli et al., 2017) and during the seasonal recession, the streamflow generated by shallow groundwater in the hillslopes and of alluvial or talus groundwater systems throughout the year (Hayashi, 2020). These explorations provide key transferable insights into the value of these four tracers for hydrologic process investigation that are relevant for comparable catchments.

## 2    Data and Methods

### 2.1    Study area

The following case study description is largely based on the paper by Michelon et al. (2021b). Vallon de Nant is a 13.4 km² headwater catchment located in the western Swiss Alps (Figure 1), with elevation ranging from 1200 to 3051 masl (mean 2012 masl). The catchment has an elongated shape and runs from south to north, along the river Avançon de Nant.

The Vallon de Nant is of national importance in Switzerland for its biodiversity (Cherix and Vittoz, 2009) and has been protected since 1969 (Natural Reserve of the Muveran). The site has been the focus of a number of recent research projects, in disciplines such as hydrology (Beria, 2020; Michelon, 2022), hydrogeology (Thornton et al., 2021), pedology (Rowley et al., 2018) biogeochemical cycling (Grand et al., 2016), geomorphology (Lane et al., 2016) and vegetation ecology (Vittoz, 2012; Giaccone et al., 2019), as well as interactions between biology and hydrology (Mächler et al., 2021) and stream ecology (Horgby et al., 2019).

The catchment lies on the backside of the Morcles nappe (Huggenberger, 1985). The Cretaceous and Tertiary lithologies are organized as a succession of thick, blocky layers exposed throughout the surrounding valley. They rest on a substratum of flysch, i.e. softer rocks (schistose marls and sandstone benches), which explains the deepening and widening of the valley at its southern part (Badoux, 1991).

In the southern part of the valley, there is a glacier (Glacier des Martinets), with a surface of 0.58 km² in 2016 (Linsbauer et al., 2021) at relatively low elevation (2126 to 2685 masl) lying on the northern, shady side of the Dent de Morcles. Due to its small size, its high debris cover and low radiation exposure, the glacier is likely to have a small contribution to the catchment-

scale streamflow (Mächler et al., 2021). The water flow paths through and below the debris-covered glacier are unknown to date and are not specifically investigated as part of the present research.

The eastern side of the catchment is marked by steep and rocky slopes associated with thin soils and debris cones at the foot of the rock walls in the north-eastern part. Along the rock walls, all lateral tributaries are ephemeral, flowing principally during the snowmelt season or shortly after the rainfall events; their extent fluctuates and is not known precisely.

The western side of the valley is associated with grassy slopes, relatively well-developed soils, and hence relatively high water-storage capacities (see pictures, Figure S1). The valley has had a relatively stable vegetation cover, composed of grassland and spruce (based on a comparison of historical with recent photographs, see Figure S2 in Supplementary Material). The distribution of stands of spruce (Dutoit, 1983), which are intermixed with corridors of scrub vegetation on the north-western slopes, are controlled by regular avalanches. Above the spruce stands, there is a transition band to subalpine and high elevation vegetation, consisting of intermixed larch, pasture, and alder.

The location of springs correlates with low slopes (see Figures S4 and S6 in Supplementary material), a topographic particularity explaining the location of springs along the right bank of the main stream and within the grassy slopes in the western area of the catchment, where the slopes are low. In the same way, the absence of tributaries over the north-western parts of the catchment are related to steep slopes, explained by the large hydraulic conductivity and locally well-developed soils. The high water-storage capacity of the well-developed soil is also indicated by salt gauging that took place along the main stream during the late summer and autumn streamflow recession period in 2016 and 2017 (see Figure S5 in Supplementary Material).

The riparian wetland (Figure 1), at least in its southern part, is made of coarse and permeable alluvial sediments associated with a high hydraulic conductivity; it could be "hydrologically active" to its full depth, which can exceed 80 m (Thornton, 2020). The extent of the stream network is based on observations during dry and wet periods (Michelon et al., 2021b) and its exact path is calculated using the Swiss digital elevation model at a resolution of 2 m (Swissalti3d, 2012). During wet periods of the study period, the stream network (as shown in Figure 1) extended to 6 km; during dry periods, the main stream contracted to as short as 2.95 km, corresponding to the channelized flow starting at 1480 masl. During the period of snow accumulation, the stream network extent was difficult to establish but there was no known occurrence of zero flow.

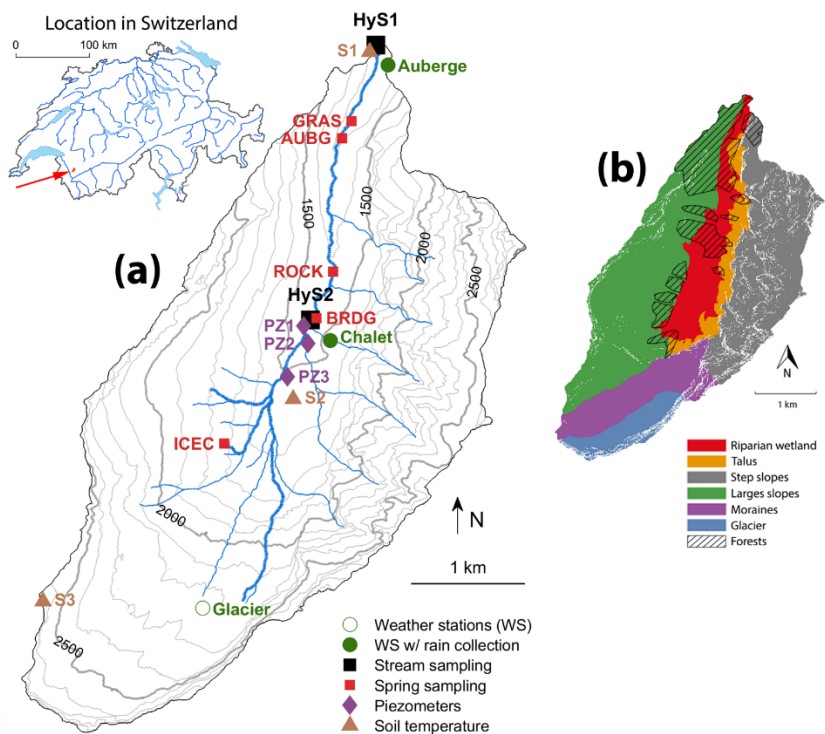

Figure 1. The map A), based on the 2m digital elevation model (Swissalti3d, 2012) shows the location of the sampling sites in the Vallon de Nant (outlet at 46.25301°N, 7.10954°E in WGS84, elevation 1200 masl). The list of locations is given in the Supplementary Material, Table S1. The stream network shows the full extent, which is only during snowmelt periods. The AUBG spring location shows where the water is sourced from, even though it is sampled from a pipe at the Auberge weather station point, 800 m further north. The map B) identifies the dominant hydrological units of the Vallon de Nant and the hatched area corresponds to forested areas.

## 2.2 Meteorological and hydrological data

Meteorological variables were monitored at three locations (Michelon et al., 2021a) along a north/south transect (at 1253 masl, 1530 masl and 2136 masl) starting in September 2016 (see Figure 1). The precipitation intensity was measured using a 24 GHz Doppler radar sensor (Lufft WS400-UMB, G. Lufft Mess- und Regeltechnik GmbH, Fellbach, Switzerland) that distinguished the precipitation phase (rain and snow), available from Michelon et al. (2021a). We used piezometric data from the work of Thornton et al. (2021) at two locations in the alluvial flood plain (Figure 1) to characterize the dynamics of the corresponding groundwater system. The streamflow at the catchment outlet (HyS1, Figure 1) was monitored since September 2015 via river height using an optical height gauge (VEGAPULS WL-61 optical height gauge, VEGA, Schiltach, Germany, see photo in Figure S3) above the middle point of a trapezoid shape weir. It averaged water height every minute continuously. The height was then converted into streamflow using a rating curve based on 55 salt gauging of discharge (Ceperley et al., 2018). The rating curve relating height to discharge is a power-relationship using the nonlinear least squares method (The Mathworks,

2017) with the trust region algorithm and least absolute residual method to obtain a 95% confidence interval. To guide the analysis of the streamflow response throughout the year, we analyzed the different streamflow periods in detail (Section3.1) based on the 7-day moving average streamflow data ($Q_{m7}$) and the daily change of $Q_{m7}$, called $\Delta Q_{m7}$. This moving average does not introduce a temporal shift since the seven-day window extends 3.5 days forward and 3.5 days backward, however it does reduce the peak flow. Its interpretation alongside the high-resolution streamflow data does not introduce a bias but rather highlights the slower processes.

## 2.3    Stable isotope composition of water

### 2.3.1    Water sampling

Water (from streams, springs, and piezometers) was either sampled manually (grab samples) or via automatic samplers placed at the outlet and an upstream location along the stream (HyS1 and HyS2; Figure 1) for stable isotope analysis ($\delta^2$H, $\delta^{17}$O, and $\delta^{18}$O). A three- or four-letter code was adopted for all sampling locations as shown on the map (Piezometers: PZ1, PZ2, PZ3; Springs: GRAS, AUBG, ROCK, BRDG, ICEC; and Stream: HyS1, HyS2; Figure 1). Twelve mL amber borosilicate glass vials with polypropylene screw-top caps with PTFE-lined silicone septa were used for all sample transport and storage. When possible, samples were stored sealed in a refrigerator (~ 4°C) until analysis but in some cases were stored at ambient temperatures. In all cases, they were kept out of direct light and heating. Automatic sampling was performed with an ISCO 6712C Compact Portable Sampler with 24 bottles of 500 mL capacity at HyS1 and an ISCO 6712 full-size portable sampler with 24 bottles of 1L capacity at HyS2 (Lincoln, Nebraska, USA). Automatic samplers were programmed to sample at 6-hour intervals over one week. The automatic sampler was programmed to fill bottles to half of their capacity, 250 mL, and 500 mL, respectively, to optimize energy usage and to prevent sample loss due to freezing, while still sampling enough water to limit fractionation due to evaporation.

A sub-sample of water was then taken manually from each bottle either in the laboratory or in the field. Original installation involved the use of pipettes and tubes inside the autosampler bottles similar to those described by von Freyberg et al. (2020), however after some experimentation and due to the Alpine and shaded microclimate of the location, fractionation due to evaporation was deemed minimal and additional components resulted in more contamination and less sampling capacity. When they froze between collection and subsampling, the bottles were closed with a cap and moved to a warmer place until the ice fully melted, and they could be subsampled.

Samples of rainfall were collected at the *Auberge* and *Chalet* meteorological stations (Figure 1) with a 13 cm diameter plastic funnel, connected to an insulated 2.5 L screw-top bag made of 147 μm PET/NY/LDPE plastic (DaklaPack, Perpignan, France), enclosed in a plastic box. The collected water was well mixed, weighed, and sub-sampled once or twice a week from May to November (outside the snowfall period).

Groundwater was sampled directly from the piezometers (Thornton et al., 2021). Prior to water sampling, the piezometers were emptied using a Geotech Peristaltic Pump (Geotech Environmental Equipment, Inc, Denver, Colorado, U.S.A.); and the freshly recharged water was sampled with the same pump and stored in 12 mL amber glass vials.

During winter 2017 and winter 2018, snow samples were collected regularly at two locations. Two different sampling methods were used: i) if distinguishable snow layers were present (visual and textural distinction) each of them was sampled individually; otherwise, ii) a single bulk sample of approximately sampled the entire profile was taken.

Snow was sealed in alimentary 700 mL zip bags made of 120 μm BOPP/LPDE plastic (DaklaPack, Perpignan, France) after evacuating as much air as possible. The collected snow samples were melted at ambient air temperature (the influence of water
vapor from air on the isotopic composition of the sample is discussed in the Appendix 1). A sub-sample of well-mixed, melted snow was taken manually in the lab from each bag.

The isotopic composition of the entire snowpack at a given snow pit was obtained with a weighted average of the values of each sampled layer according to depth, as an approximation for the equivalent bulk isotope composition assuming a uniform density.

### 2.3.2    Analysis of the isotopic composition of water

Stable isotope compositions of water, expressed using the δ notation ($\delta^2H$, $\delta^{17}O$ and $\delta^{18}O$), were measured with a Picarro 2140-i Wavelength-Scanned Cavity Ring Down Spectrometer (Picarro Inc., Santa Clara, California, U.S.A.), using 2.0 mL glass vials closed with screw-top caps with silicone Rubber/TPFE septa and filled with 1.8 mL of filtered water. Samples were injected between 6 and 8 times (Penna et al., 2012). The first 3 injections were discarded to avoid memory effects (Penna et
al., 2012). The raw values were then corrected according to a standard curve determined with 3 internal standards, which are regularly calibrated against the international standards of VSMOW (Vienna Standard Mean Ocean Water) and SLAP (Standard Light Antarctic Precipitation) of the IAEA (International Atomic Energy Agency)(Coplen, 1994). Each standard was injected 12 to 15 times, and data from the final 6 injections were kept. Delta units of isotope compositions (Coplen, 1994) are reported in per mil and the strategy used for the analysis is similar to the one described in the work of Schauer et al. (2016). The median
analytical errors obtained with this method are 0.4 ‰ for $\delta^2H$, 0.01 ‰ for $\delta^{17}O$, 0.04 ‰ for $\delta^{18}O$.

Based on these measures, we compute d-excess (Dansgaard, 1964) and $^{17}O$-excess (Barkan and Luz, 2005; Landais et al., 2006):

d-excess: $= \delta^2H - 8 \cdot \delta^{18}O,$ (1)

$^{17}$O-excess: $= 10^6 \left( ln\left(\frac{\delta^{17}O}{1000} + 1\right) - \lambda_{ref} \cdot ln\left(\frac{\delta^{18}O}{1000} + 1\right) \right),$ (2)

with $\lambda_{ref} = 0.528$ (Meijer and Li, 1998; Barkan and Luz, 2005; Landais et al., 2008). From regression of $ln(\delta^{17}O/1000 + 1)$ against $ln(\delta^{18}O/1000 + 1)$, we obtain a similar slope for our samples ($\lambda_{ref} = 0.528$), which confirms the universality of this value. However, memory effects can notably influence the $\delta^{17}O$ measurements in cases of larger variations in values within any one sequence measured (Vallet-Coulomb et al., 2021).

In order to gain insights into local evaporative processes, we compute the line-conditioned excess lc-excess (Landwehr and Coplen, 2006) based on our local meteoric water line LMWL :

$$\text{lc-excess:} = \delta^2 H - a \cdot \delta^{18} O - b. \tag{3}$$

### 2.3.3 Isotopic lapse rate estimation

We estimated elevation gradients of isotopic ratios, i.e., lapse rates, based on the median of measurements at different locations for precipitation and streamflow. For precipitation, we used the measurements at Auberge station (elevation 1253 masl) and Chalet station (elevation 1517 masl). For streamflow, we used the measurements of station HyS1 (elevation 1248 masl) and HyS2 (elevation 1469 masl). For other sampled waters, we did not establish lapse rates due to a varying number of samples and inconsistent sampling dates.

## 2.4 Water temperature and electrical conductivity measurements

The water temperature of four springs was recorded every 30 minutes (every 15 minutes for GRAS and ROCK springs) with Hobo temperature loggers (Onset Computer Corporation, Bourne, MA, U.S.A.) for periods lasting between 12 and 21 months. Based on these recordings, we estimated lag times with respect to air temperature and diel and annual amplitudes. The original time resolution of 1 minute for the stream, 30 minutes for piezometers (PZ) and 2 minutes for springs was kept for the diel temperature maximum amplitude but aggregated to 1 day to compute the annual temperature maximum amplitude (using a 7-day moving average, see Figure S9). Lag times were obtained by maximizing cross correlation between the 1-day signal and the one for the reference air temperature signal (Figure 1, Auberge station). Electrical conductivity (EC) was measured for all collected water samples except snowpack, either directly in the field with a WTW Multi 3510 IDS connected to a WTW TetraCon 925 probe (Xylem Analytics Germany Sales GmbH & Co, Weilheim, Germany) or in the laboratory directly in the vials using a JENWAY 4510 Conductivity Meter with a 6 mm glass probe (Stone, U.K.). Comparison of duplicate measurements using both probes (compensated in temperature) demonstrated a correlation coefficient of $R^2=0.89$ despite a delay of 23 to 30 months between the *in situ* and laboratory measurements (see Figure S7 in Supplementary Material).

Water temperature was analyzed with a simple analytical temperature model with sinusoidal initial conditions (e.g. Elias et al., 2004) to compute a rough depth that corresponded to such a lag, $L$ (for details see Appendix 2). For example, if we assume that the thermal diffusivity of soil is a typical value of $5.56 \ 10^{-7}$ m$^2$/s (Elias et al., 2004), a lag of 41 days corresponds to a depth of 1.7 m, where as a lag of 39 days corresponds to a depth 1.6 m.

## 2.5 Soil temperature measurements

The soil temperature at three different elevations (at 1240 m, 1530 m and 2640 masl, see Figure 1) was monitored from July 2009 to November 2018 using GeoPrecision M-Log5W (GeoPrecision GmbH, Ettlingen, Germany) at 10 cm depth (see Figure 1) and recorded hourly (Vittoz, 2021). Although it is common to study the effect of snow cover on near surface soil temperature

especially for prediction of microclimates and habitats (Rixen et al., 2008; Freppaz et al., 2018; Giaccone et al., 2019), the inverse focus is also relevant and soil temperature is a good proxy for snow cover (Bender et al., 2020; Staub et al., 2015), making distributed observations of soil temperature particularly useful for observing the expansion and contraction of snow cover. Strong diel variations of can be associated with snow-free soils, which correlate with the larger amplitude air temperature fluctuations and radiative exchanges.

## 3    Results

### 3.1    Streamflow response characterization

The annual average streamflow over 2017 and 2019 was between 0.46 to 0.62 $m^3$ $s^{-1}$ (3.0 to 4.0 mm $d^{-1}$) but fluctuated between 0.02 to 0.03 $m^3$ $s^{-1}$ (0.12 to 0.18 mm $d^{-1}$) and 2.4 to 3.1 $m^3$ $s^{-1}$ (15.5 to 19.7 mm $d^{-1}$) as shown in Figure 2. Peak monthly flow occurred either in June (for 2017-2018 and 2018-2019) or July (2016-2017, the year with abundant winter snow). Flood events resulted in streamflow peaks as high as 5.8 $m^3$ $s^{-1}$ and up to 7.2 $m^3$ $s^{-1}$ (37.4 to 46.3 mm $d^{-1}$) over 1 hour and from 6.9 to 8.5 $m^3$ $s^{-1}$ (44.4 to 54.6 mm $d^{-1}$) over 10 minutes. The mean temperature of the streamflow at the outlet (1200 masl) was 5.0 °C and ranged from 0 °C when the river was frozen during some winter periods to a daily temperature of 10.0 °C during summer. As a comparison, the annual mean air temperature at mean elevation (2012 masl) was 3.1 °C in 2017 (considerable data gaps in 2018), based on data from 3 stations (Glacier, Auberge, Chalet; Fig. 1).

Based on the moving average streamflow, $Q_{m7}$, and the corresponding daily fluctuations, $\Delta Q_{m7}$, we identified four characteristic streamflow periods (Figure 3, Figure S9 and Table S2 in the supplement): Baseflow (B); Early melt (E); Melt (M); And recession (R). The two main features of these periods are that streamflow increased during E and M and decreased during R and B and that there was a range of daily values that was very low during B (around 0.02 $m^3$/s), relatively low during E and R (0.1 $m^3$/s) and considerably higher during M (0.3 $m^3$/s).

Period B extended from the end of September to early spring (between mid-March and beginning of April), when the streamflow was approximately 1 mm $d^{-1}$, which is typical for catchments at comparable elevations (Floriancic et al., 2018). The baseflow exhibited a very slow decrease across the period B, with almost no diel variations even though some streamflow peaks occured due to exceptional rainfall events or warm periods (e.g., January 2018). B had $Q_{m7}$ values lower than the 30th percentile of observed streamflow.

During E, streamflow started increasing to a few mm $d^{-1}$, preceding the main snow melt period M, but the daily range did not show a large increase. E lasted up to several weeks in certain years (e.g., in 2017) and was absent in one year (2018), when warming, and thus melt, occurred extremely quickly. E had $Q_{m7}$ values around the 50[th] percentile. The average $\Delta Q_{m7}$ was 0.64 mm per day (0.1 $m^3$/s). M corresponded to the period when snowmelt dominated the water input, which is sometimes referred to as the snowmelt freshet (Frisbee et al., 2011). Compared to E, there was a much higher diel variation in streamflow (resulting from diel snowmelt patterns). In 2017, M started at the beginning of May, and it started at least a month earlier in 2018. M had

$Q_{m7}$ values above the 80[th] percentile. The average $\Delta Q_{m7}$ was greater than 0.64 mm per day (0.1 m³/s). The time of diel peak discharge (in absence of rainfall input) was between mid-day or late afternoon.

R was set to begin after the annual maximum of average $Q_{m7}$, which in 2016 was the end of June (preceded by a very snow-rich winter), in 2017, was the end of May and in 2018, shifted to beginning of June. R results from a combination of reduced input from snowmelt and evaporation, as clearly visible in the significant diel streamflow variations during R. $Q_{m7}$ values were

between the 70 and 75[th] percentiles, $\Delta Q_{m7}$ was close to -0.64 mm per day (-0.1 m³/s), and the time of diel peak streamflow was in the morning or early afternoon. A summary of streamflow characteristics during all four periods is included in Table S2.

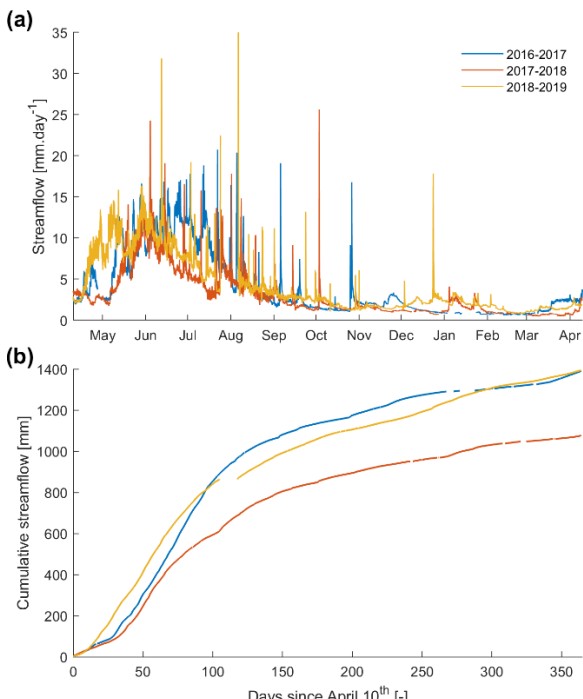

Figure 2. Comparison of the streamflow evolution (a) and cumulation (b) per year for 3 years. The start is here 10 April of
each year, which corresponds to the start of the earliest start of the early melt period E (Section 3.1). There are gaps in the data, 6.4%, 3.2% and 4.6% of the time series are missing (respectively in 2016-2017, 2017-2018 and 2018-2019).

### 3.2    Soil temperature

#### 3.2.1    Soil temperature dynamics

The fluctuations in soil temperature at three different elevations (1240 m, 1530 m and 2640 masl) revealed seasonal, daily and
spatial variation (Figure 3). As winter approached each year, soil temperature gradually dropped towards, but did not reach 0°C. The slightly positive temperature can be explained by ground heat flux. Once snow fell, the insulation it provided was visible as a dampened diel temperature variation that was less correlated with daily air temperature fluctuations than in snow free periods. Some isolated temperature spikes were observed during winter, probably due to rain-on-snow events (e.g. the

spike during winter 2016 in the green line in Figure 3, at the lowest elevation) that were visible in the temperature signal but
apparently did not generate any streamflow. Unfortunately, no other observed tracers were available during these periods to
confirm this hypothesis.

The negative temperatures measured during the 2016-2017 winter period by two soil temperature probes (at 1530 m and 2640
masl) can be explained by the cold air temperature and the exceptionally dry winter with a low snow cover. The variation in
temperature at the three different elevations in Figure 3 displays the elevation gradient that alters the start of the snow-free
period at the different elevations and the elevation-dependency of the timing of strong warming (of more than 5 °C) between
March and July. The start of the snow-free period occurred between 4 (2018) and 8 weeks (2017) earlier at 1240 masl elevation
than at 1530 masl, whereas between 1530 masl and 2640 masl, it was delayed by between 3.5 (2018) and 8 weeks (2016).

Similarly, the soil temperature time series clearly showed the elevation dependency of the arrival of snow. It arrived much
earlier (12 weeks) in autumn 2016 at the highest elevation as compared to the two lower elevations. In 2017, the seasonal snow
cover onset occurred at a similar time at all elevations, as we see that all diel temperature variation disappears between October
22$^{nd}$ and November 25$^{th}$. A summary of snow-free dates as extracted from the temperature recordings is available in the
Supplementary Material (Table S3).

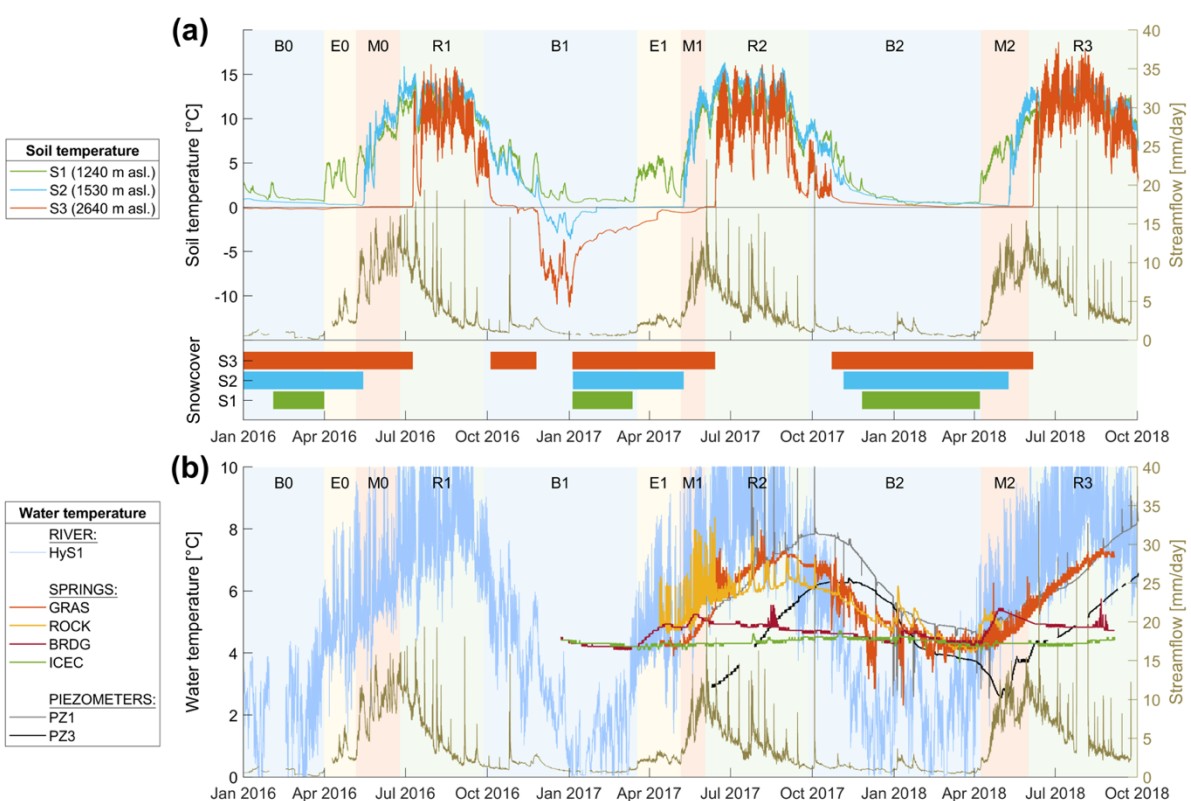

Figure 3. Evolution of soil temperature (a) at 3 locations within the Vallon de Nant catchment (S1, S2, S3) and (b) of the
stream at the outlet (HyS1), 4 springs (GRAS, ROCK, BRDG, ICEC) and groundwater at 2 locations (PZ1, PZ3) between

January 2016 and October 2018. A summary of dates corresponding to the 11 runoff periods divided by baseflow (B0, B1, B2), early melt (E0, E1), melt (M0, M1, M2) and recession (R1, R2, R3) is available in the Supplementary Material (Table S2).

### 3.2.2 Soil temperature links to streamflow

Interesting features were observed because soil temperature was examined with respect to the identified streamflow periods. For all three melt periods (M), soil temperature recordings from the highest elevation showed the presence of snow until the start of the recession period (R), which demonstrated the late melt of seasonal snow in some areas of this catchment. The start of the two early melt periods (E) in 2016 and 2017 corresponded to the disappearance of snow at the lowest soil temperature measurement point (1230 masl).

Decline in soil temperature started at a similar time across all elevations and occurred close to the start of the streamflow baseflow periods (B), or in other words, the significant decrease in soil temperature only started when the streamflow recession period (R) was already advanced. In the winter of 2016/2017, winter streamflow fluctuations were reflected in soil temperature, whereas the mid-winter streamflow rise in January 2018 was not visible in any of the soil temperature recordings, however this might be due to errors in recording river stage caused e.g., by accumulated sediment (Michelon, 2022, chapter 3).

Finally, from the covariation between soil temperature and streamflow, we can deduce that during M, runoff-generating rainfall events coincide with cold spells, whereas during autumn, runoff-generating rainfall events coincide with warm spells (e.g., October 2016 and 2017).

## 3.3 Water temperature

### 3.3.1 Influence of air temperature on stream temperature

Average recorded stream temperature at the outlet was 5.0 °C, which was slightly higher than 3.1 °C, the average recorded air temperature at mean elevation 2012 masl. The fluctuation of water temperature at the catchment outlet (HyS1, Figure 1) were correlated with the variations of the air temperature ($R^2 = 0.87$) at the Auberge weather station and the annual cycle showed no lag between them. The close correspondence is explained by the fact that the in-stream travel time is long enough for atmospheric heat exchange to exert a strong influence on water temperature (Gallice et al., 2015). The importance of instream atmospheric heat exchange can also explain the high annual and diel temperature amplitudes (Table 1), which corresponded closely to the observed air temperature amplitudes over the year (between 17.5 and 19.5 °C at the lower elevations, with a 30-day moving average).

### 3.3.2 Groundwater temperature patterns in springs and piezometers

Observation of patterns in the temperature of springs and groundwater reveal hints of underlying flow generation processes. Indicators of these processes include correlation between water and air temperature, diel temperature variations, temperature

response to rainfall events, the overall pattern and shape of temperature fluctuations on a seasonal scale, mean values and convergence between different points over the study area and temperature anomalies and their timing. First, we observe that
the correlation between water temperature in springs and in piezometers and air temperature, measured at the Auberge station, varied by location (Table 2); PZ1 correlated the strongest ($R^2$=0.80) and ICEC the least ($R^2$=0.56) and none as strongly as the surface water ($R^2$ = 0.87, described above).

In general, diel temperature variations were rare and can probably be explained by poor measurement when water volume was low. The temperature in some springs reacted to precipitation while that in others did not. The GRAS spring had a permanent
but small outflow of only a few liters per minute (personal observation). Since the temperature was recorded directly in the outflowing water, the sensor might have been heated up by atmospheric heat exchange when outflow was low. This most probably explains some strong sub-diel temperature fluctuations of the GRAS (and ROCK) springs (Figure 3). Despite these diel fluctuations, the GRAS temperature signal did not react to summer rainfall events (visible as peaks on the streamflow), whereas ROCK reacted.

The shape of the curve of temperature fluctuations gives us clues to the flow that fed the springs. For example, that of the BRDG spring differed from the sinusoidal shape of the GRAS and ROCK springs, which match the air temperature variations more closely. The BRDG spring signal showed a constant temperature during winter, with an increase during E and M, when it rose from 4.3°C to 5.4°C over 3 weeks at the beginning of M2. The temperature stopped rising around when the soil temperature at mid-elevation indicated snow disappearance (blue bar in Figure 3) and then receded to winter base temperature.
By combining the observation of a strong reaction during melt at low elevations with the return to a base temperature during winter, we can deduce that BRDG spring is fed by snowmelt from low elevations (from the right bank riparian area where it is located) during spring and by groundwater the rest of the year.

All spring temperatures converged to around 4.3 °C at the end of B2 (the only winter period measured in all springs), which corresponds to the almost constant temperature of ICEC spring (annual amplitude of 0.4 °C, Table 1). This may indicate that
at this point in the year, all springs are fed by a common or similar ground water source with little influence of intermediary subsurface or surface flows.

The two piezometers that access the groundwater (PZ1 and PZ3) are in the alluvial floodplain across which the stream meanders into an alluvial plane. During intense rainfall events, PZ1 shows strong positive temperature excursions, which even exceeded the temperature of the main channel in summer, however its winter anomalies were less extreme. The annual cycle
of the PZ1 temperature reached its maximum temperature of 7.9 °C with a delay of 74 days (2.5 months) after the air maximum and exceeded the maximum recorded in the springs by 1.5 °C. The strong delay of the annual cycle together with the warm temperatures and relatively small amplitude dampening compared to ROCK and GRAS springs suggests that it is influenced by a large storage volume which induces the delay and is closely connected to heat input from the surface.

PZ3 shows the same annual temperature amplitude as PZ1 but has an even longer delay (21 days with respect to PZ1) and has
a negative offset of 1.5 °C of its maxima (6.4 °C for PZ3) compared to PZ1, possibly related to the higher elevation and more

northern aspect of its source area (PZ3 is located 30 m higher, in the more north-facing part of the catchment). PZ3 has, however an average temperature of 4.8 °C closer to the one of the springs.

A distinctive feature of PZ3 is its temperature decrease during M2, in phase (but in opposite direction) with the streamflow increase. This suggests a direct, relatively important cold input during snow melt, resulting from a high hydrologic connectivity of PZ3 to snowmelt water (either directly or via exfiltration from the stream) and a low storage volume during this time of the year.

### 3.3.3 Flow path depth estimation from temperature measurements

Dampening depths estimated with the simple temperature model (Appendix 2) from groundwater temperature patterns ranged between 1.2 and 5.4 meters (Table 2) and the lag in temperature for those same points compared with streamflow ranged from 41 to 133 days. We attribute these values to the delay resulting from heat conduction (depending on the soil's thermal diffusivity $D$) and advection with water flow. The one exception is BRDG, for which lag estimation fails, perhaps indicating that this spring is not truly groundwater fed. These lag values are furthermore coherent with the dampening: stronger lags correspond to stronger dampening and are associated with deeper depths. They should however be interpreted with care as: i) the presence of an insulating snowpack on the hillslopes prevents heat advection during winter in a similar way that soil would, thereby further contributing to temperature lags and amplitude dampening in the subsurface; and ii) the model is only based on heat conduction and does not account for advection that could be locally important, particularly during snowmelt inputs.

The BRDG spring highlights these limitations, as the temperature variation over the year (0.9 °C) happens over few weeks during the melt periods (M1 and M2). This variation shows a strong reactivity to the snowmelt input, but the resulting estimation of flow path depth (0.2 m) is obviously erroneous. At this time, the maximum air temperature is not reached yet (during R2 and R3) and the expected heat signal transferred from air by conduction later in the year is not visible.

Table 1. Statistics of temperature time series recorded in the stream, piezometers and springs. The dampening depth estimated for the BRDG spring (*) is biased because of a positive anomaly of temperature due to snowmelt input (see text).

| Water source | Mean T [°C] | Max T [°C] | Annual T amplitude [°C] | Max. diel T amplitude [°C] | Cross corr. w/ air T Lag [days] | Max corr. [-] | Dampening depth [m] | Snowmelt anomaly | Rainfall anomaly |
|---|---|---|---|---|---|---|---|---|---|
| Stream | 5.0 | 13.4 | 8.8 | 11.4 | 0 | 0.92 | - | - | - |
| PZ1 | 6.3 | 7.8 | 3.8 | 4.0 (punctually) | 79 | 0.80 | 3.2 | No | Yes |
| PZ3 | 4.8 | 6.3 | 3.7 | 0.5 (punctually) | 105 | 0.68 | 4.3 | Negative | No |
| GRAS | 5.5 | 7.4 | 3.0 | 2.4 | 41 | 0.76 | 1.7 | No | Yes |
| ROCK | 5.4 | 6.7 | 2.5 | 2.9 | 39 | 0.76 | 1.6 | Positive | yes |
| BRDG | 4.7 | 5.6 | 0.9 | 0.9 | 6 | 0.68 | 0.2* | Positive | No |
| ICEC | 4.3 | 4.7 | 0.4 | 0.6 (noise) | 133 | 0.54 | 5.4 | No | No |

### 3.4 Electrical conductivity

The range of conductivity for stream, springs, groundwater, rainfall, snowpack, glacier and vegetation water samples is shown in Figure 4. The time series of conductivity for 5 springs is shown in Figure 5, at the outlet HyS1 and at the upper subcatchment outlet, HyS2, in Figure 6 and for rainfall (from Auberge and Chalet weather stations) in Figure 7.

The median electrical conductivity of 216 µS/cm in streamflow at the outlet is relatively high for alpine environments. The electrical conductivity observed at the outlet and in the upper subcatchment (HyS2, median EC of 215 µS/cm) were very close. Assuming a spatial homogeneity between flow path depth and flow velocity, their proximity suggests a similar flow path length distribution. The temporal evolution of stream EC included a decrease in EC during the melt season. A time lag between seasonal cycles in EC and streamflow cycles is seen in Figure 6. This event-scale lag accumulates into a shift of the seasonal

cycle in streamflow and EC.

All springs, except ICEC had higher EC values than the stream or the directly sampled groundwater. Higher EC values point towards longer flow paths in the subsurface, either vertically or laterally (Cano-Paoli et al., 2019), or alternatively longer residence times of the water, hence lower flow rates. The spring with the highest EC (GRAS, median EC of 271 µS/cm) showed the least temperature dampening and the spring with the lowest EC (ICEC, median 211 µS/cm) shows the most

dampening (where high amounts of dampening indicates deep flow paths in the subsurface). Assuming a homogeneous underlying geology, the only possible explanation of EC signals in conjunction with the temperature signals is that low EC values of subsurface water result from short flow paths in the shallow subsurface (GRAS spring) and relatively high EC values result from longer and deep flow paths (ICEC).

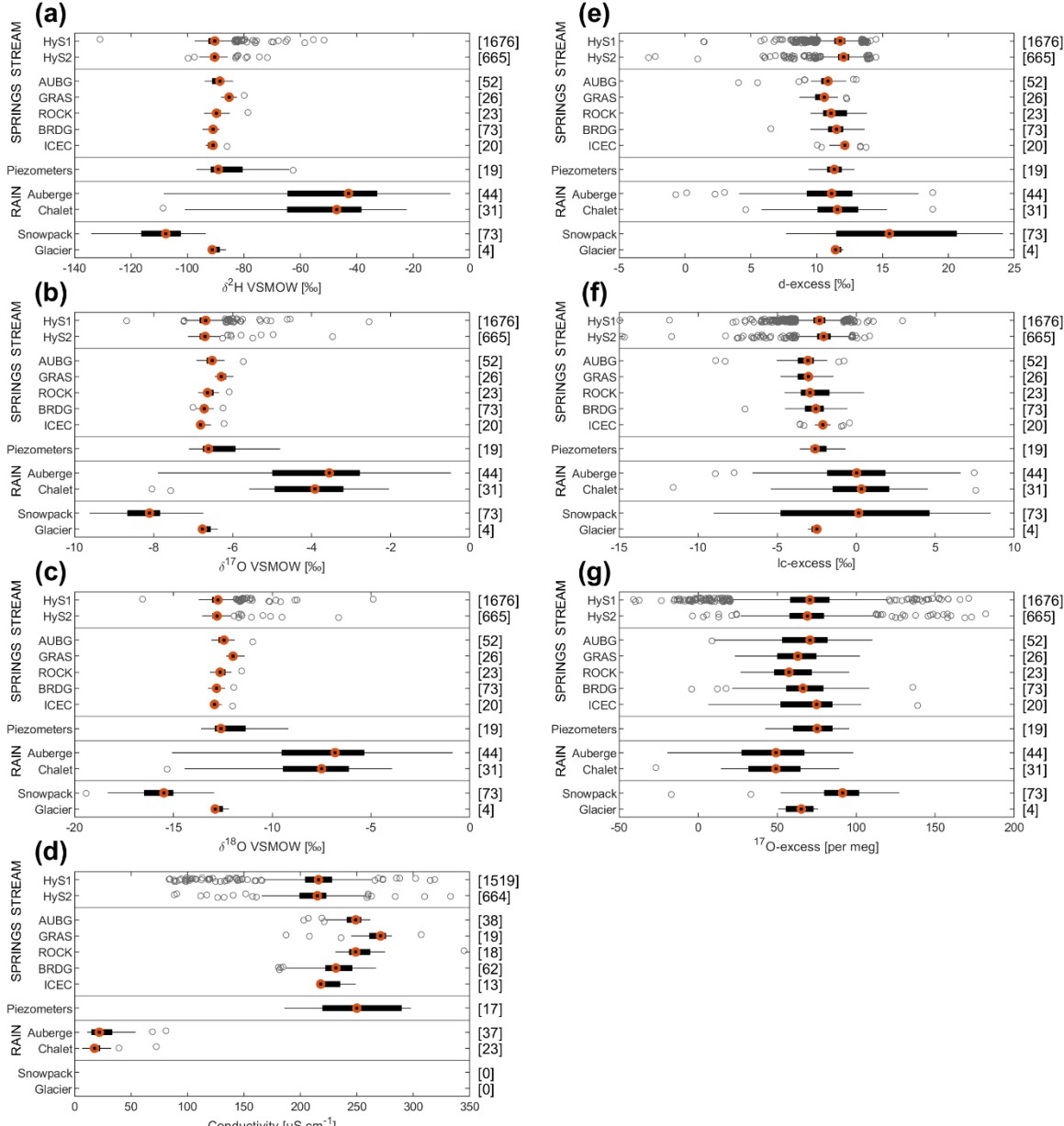

Figure 4. Range of $\delta^2H$, $\delta^{17}O$, $\delta^{18}O$, conductivity, d-excess, lc-excess and $^{17}O$-excess for stream, springs, groundwater, rainfall, snowpack, glacier, and vegetation water samples. For measurements of the same water type (STREAM, SPRINGS, RAIN), the lowest sampling elevation is given on top, the highest on the bottom. The left and right end of each box show the 25th and 75th percentiles and the middle point is the median. The whiskers go up to 1.5 times the interquartile range; values beyond the whiskers (outliers) are marked with circles. The values on the right y-axis of the figures are the number of samples in each category. Note that there are no conductivity measurements

for snowpack and vegetation water samples. For $^{17}O$-excess the values of vegetations samples are out of the box (median 572 ‰, 25th and 75th percentiles are 374 ‰ and 742‰, resp., and whiskers are from 31 ‰ to 8329 ‰).

## 3.5    Stable isotope composition of water

The range of $\delta^2$H, $\delta^{17}$O, $\delta^{18}$O, d-excess, lc-excess and $^{17}$O-excess for stream, springs, groundwater, rainfall, snowpack, glacier and vegetation water samples is shown in Figure 4. The time series of $\delta^{18}$O, $\delta^{17}$O , d-excess, lc-excess and $^{17}$O-excess for 5
springs is shown in Figure 5, at the outlet HyS1 and at the upper subcatchment outlet, HyS2, in Figure 6, and for rainfall (from Auberge and Chalet weather stations) and snowpack in Figure 7. Additional figures displaying further variables (i.e. $\delta^2$H) are in the Supplement (Figure S10).

### 3.5.1    Ranges and lapse rates of $\delta^2$H, $\delta^{17}$O and $\delta^{18}$O

The median $\delta^{18}$O value of all streamflow samples was -12.7 ‰. An isotopic lapse rate of 0.84 ‰/100 m for $\delta^2$H and 0.13 ‰/100 m for $\delta^{18}$O was computed from our precipitation water samples between the Auberge and Chalet stations (with higher median value at the lower Auberge weather station), which is approximately half the isotopic lapse rate of precipitation observed in Switzerland (e.g. Beria et al., 2018). However, this lapse rate was not seen in stream water isotopes (Figure 4 a-c) measured at HyS1 (1248 masl) and HyS2 (1478 masl). The distribution of elevations feeding HyS1 (mean elevation 2165
masl) was most probably not sufficiently different from the distribution feeding HyS2 (mean elevation 2196 masl) to result in a significant shift of the isotopic values at the two streamflow sampling locations, despite the isotopic lapse rate in precipitation. The median isotopic values for the sampled springs showed an elevation gradient (Figure 4a-c, showing the spring values in order of elevation from AUBG to ICEC), except the GRAS spring, which showed a significantly higher median isotopic value. This suggests that GRAS spring might receive water only from a small low elevation sub-catchment and not from the high
rock walls located next to it.

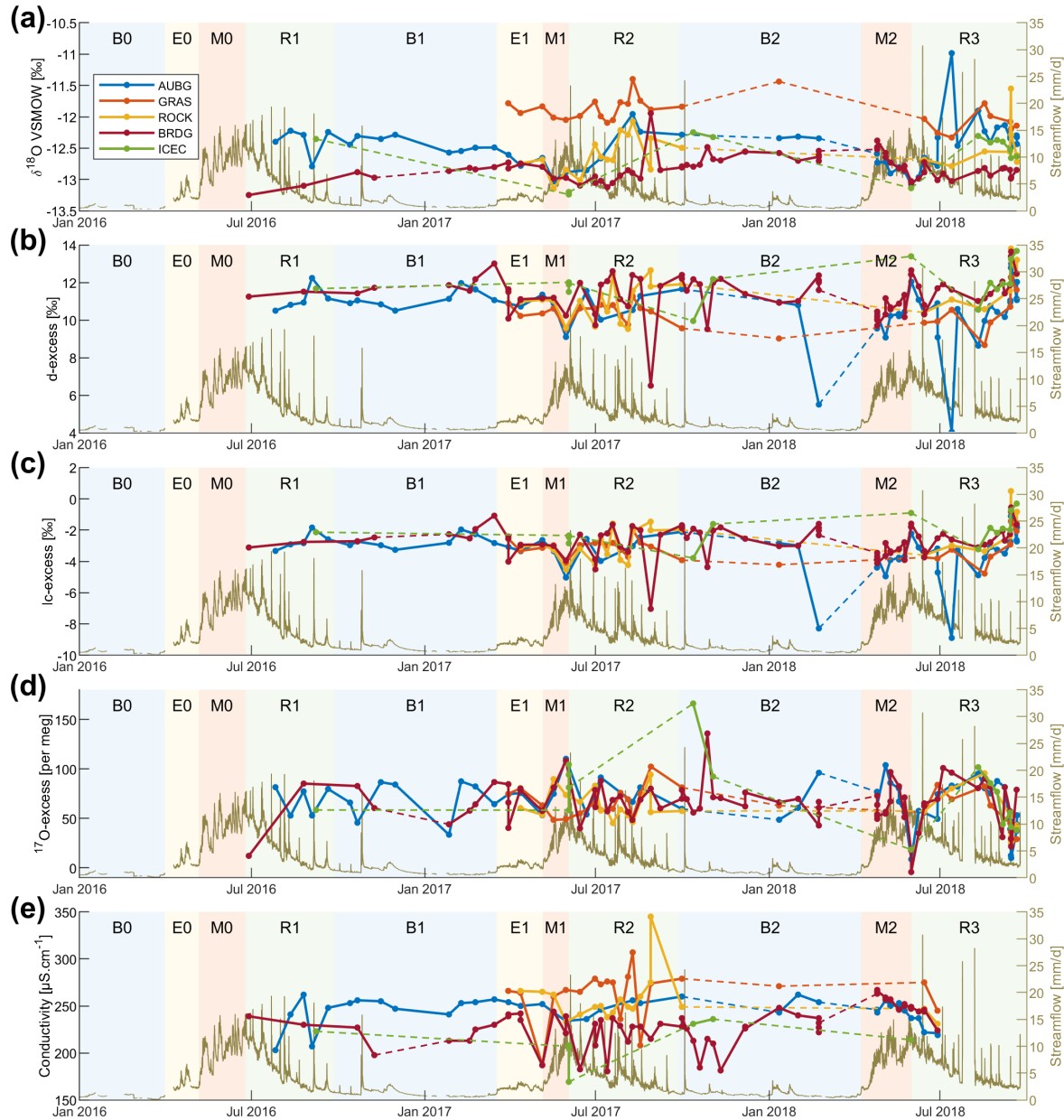

Figure 5. Time series of $\delta^2H$, $\delta^{17}O$, $\delta^{18}O$, conductivity, d-excess, lc-excess and $^{17}O$-excess for 5 springs (location on map Figure 1 A). We connect the dots to facilitate visualization since there are 5 separate springs traced.

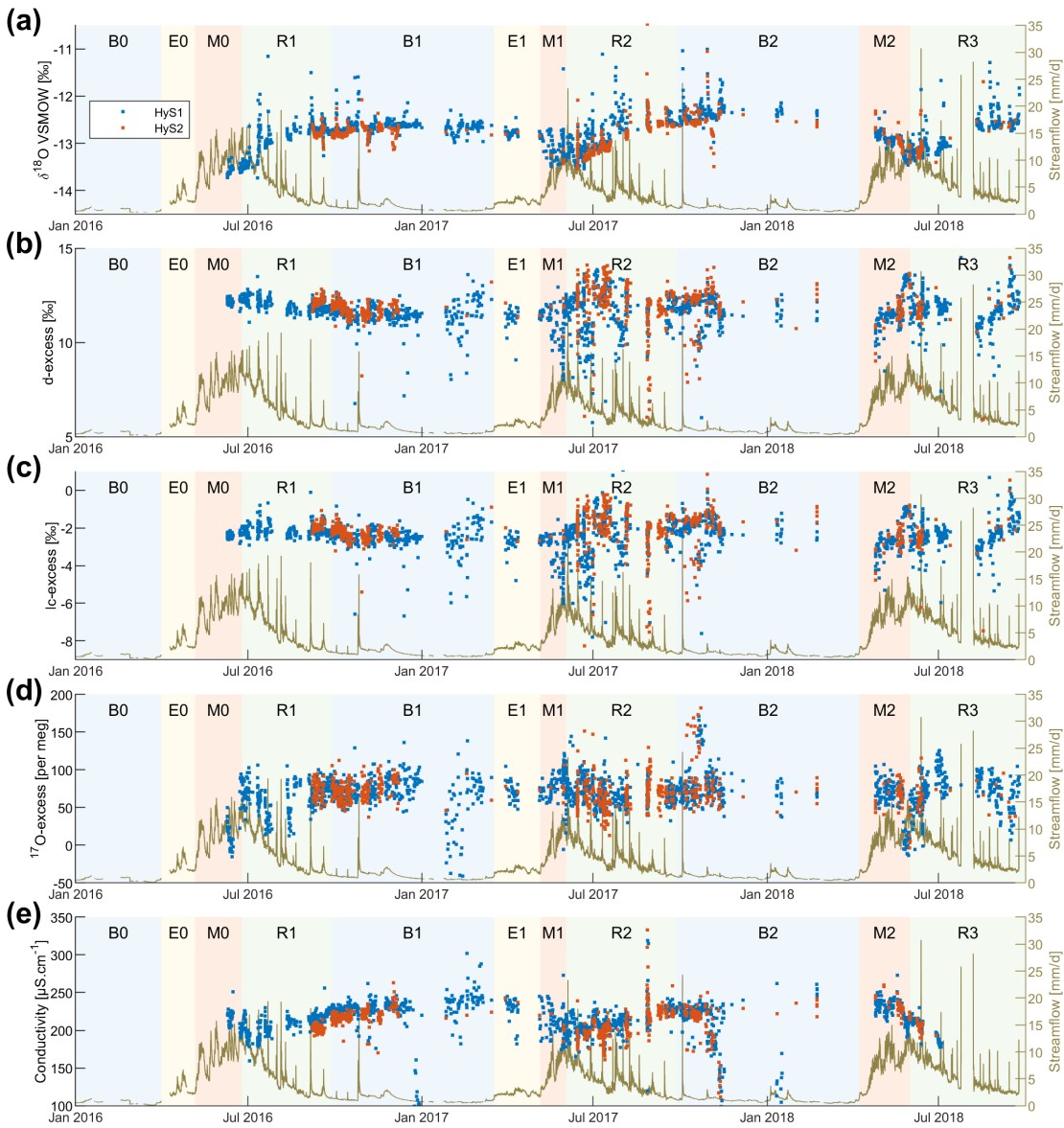

Figure 6. Time series of δ²H, δ¹⁷O, δ¹⁸O, conductivity, d-excess, lc-excess and ¹⁷O-excess at the Vallon de Nant outlet HyS1 and at the upper subcatchment HyS2 (location on map Figure 1 A).

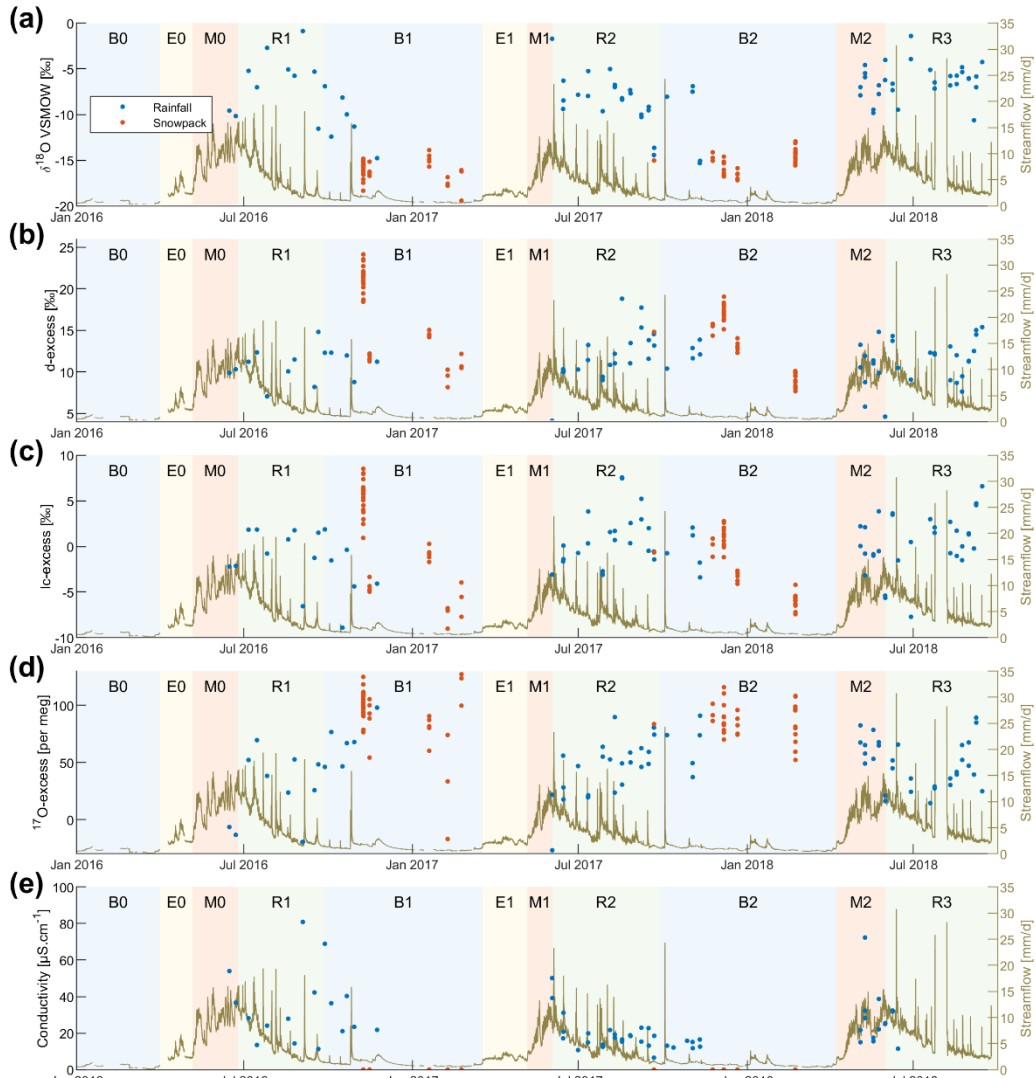


Figure 7. Time series of $\delta^2H$, $\delta^{17}O$, $\delta^{18}O$, conductivity, d-excess, lc-excess and $^{17}O$-excess for rainfall (from Auberge and Chalet weather stations, location on map Figure 1 A) and snowpack. Note that the conductivity of snowpack has not been measured.

### 3.5.2   Dynamics of $\delta^2H$, $\delta^{17}O$ and $\delta^{18}O$ in springs

The fluctuations of the isotopic composition of the 6 monitored springs between July 2016 and September 2018 is discussed qualitatively based on the streamflow periods (see Figure 5). Because the variations in values of $\delta^2H$, $\delta^{17}O$ and $\delta^{18}O$, correlate well, we will only discuss $\delta^{18}O$. Corresponding figures of the other isotopic values can be found in the supplement.

Despite some variability, the AUBG spring $\delta^{18}O$ values remained relatively constant ($\delta^{18}O$ between -12.8 ‰ and -12.2 ‰) during the 2016 streamflow recession R1 and then slowly decreased throughout the 2016/2017 baseflow period B1. Meanwhile, the BRDG spring started with lower isotopic values (-13.3 ‰) but got enriched in heavier isotopes through R1 and B1 to finally have a similar composition during the 2017 early melt period (E1) compared to the AUBG (and ROCK) springs, also with a subsequent decreasing trend in the heavy isotopes.

The 2017 minimum isotopic values of the AUBG, ROCK and BRDG springs were reached around the time of 2017 maximum streamflow and then diverged during the 2017 recession period (R2), increasing at a different rate: the $\delta$-values of AUBG and ROCK springs increased quickly (+1.0 ‰ in 3 months), while the BRDG spring values only showed a slow increase through the winter period (B2).

The beginning of the 2018 melt period was exceptionally fast, without an early melt period. The springs sampling started 3 weeks after the beginning of the early melt period (E), with a significant part of the snowpack having melted already. During M1, the isotopic composition of the AUBG and BRDG springs over this period showed a constant decrease in the heavy isotopes until the 2018 streamflow maximum. At the inverse of R1, the BRDG composition remained constant during the recession, while the AUBG spring increased quickly in the $\delta$-values.

The pattern repeated during the 2018 melt period (M2) with a decrease in $\delta$-values, which then diverged at different rates. Again, the BRDG spring $\delta$-values increased slowly, while for the AUBG and ROCK springs the increase was faster.

The ICEC spring, located on the western slopes (Figure 1), followed the same isotopic pattern as the AUBG spring. Although, because of its lower sampling rate, points were missing at the critical moments during the melting periods, which does not allow us to discuss the differences in timing. The ICEC spring showed higher isotopic values compared to BRDG even though it is located at a higher elevation, which can be explained by the higher maximum elevation of the mountain ridge upstream of BRDG compared to ICEC (see Figure 1), which most certainly leads to a higher snowfall contribution to BRDG as snowmelt isotopes are more depleted in heavier isotopes than rainfall (Figure 7).

As discussed earlier, the GRAS spring behaves differently from other springs, with higher $\delta$-values than all the others in 2017. EC and temperature measurements indicate that this spring has relatively shallow flow paths and its $\delta$-values also suggested a larger proportion of rainfall-derived water (which has a higher average $\delta$-values than snowmelt; Figure 7).

### 3.5.3    Dynamics of $\delta^2H$, $\delta^{17}O$ and $\delta^{18}O$ in streamflow

Because of the close overall correlation between $\delta^{18}O$ and $\delta^2H$ (Figure S10), and $\delta^{18}O$ and $\delta^{17}O$ values, we will highlight $\delta^{18}O$ in our discussion and figures (Figure 4-7). The temporal evolution of the isotopic values in the streamflow showed high $\delta^{18}O$ (and $\delta^2H$) during winter baseflow, close to the median value of all sampled subsurface water bodies, and a significant decrease in the isotopic composition during the melt periods. Streamflow is thus largely fed by recent (isotopically light) snowmelt during the melt period; the decrease of the $\delta^{18}O$ (or $\delta^2H$) is proportional to the amount of snowmelt, with a larger decrease in 2017 compared to 2018. The $\delta^{18}O$ (and $\delta^2H$) value did not decrease during E.

### 3.5.4    Excess (d- and lc-)

Rainfall had a median d-excess value of 11.3 ‰ and snowpack of 15.5 ‰. However, as we did not systematically sample fresh snowfall and snowpack separately, it was hard to estimate the extent of snow sublimation in Vallon de Nant.

Surface and subsurface water samples showed median d-excess values close to that of rainfall and considerably lower than the median value for snow. The apparent surface and subsurface water samples bias towards the d-excess value of rain can be explained by secondary evaporation (e.g., from soil); the soil water that remains (and that ultimately recharges groundwater and the streams) thus has a lower d-excess value than either rainfall or meltwater. Thus, the d-excess could be interpreted as the "evaporative exposure" of water during the time since precipitation, but it is rather difficult to interpret in terms of local scale process information. Even if the difference in median value between rainfall and snowfall is significant enough for a separation, because of the potential for transformation along the flow path, it will not be as valuable as the $\delta^{18}O$ (and $\delta^2H$) values directly. For ice melt, d-excess values are too close to those of rainfall for providing further insights into its importance in streamflow.

The LMWL was calculated using linear regression between $\delta^{18}O$ and $\delta^2H$ of 75 rainfall samples with a slope of 7.38 and an intercept of 6.15 (see Figure S10). The median analytical error was 0.4 ‰ for d-excess and lc-excess, and 8 per meg for $^{17}O$-excess. Compared with d-excess, LC-excess for rainfall samples indicated the spread of precipitation isotopes around the LMWL (Figure 4F). Our LMWL deviated from the GMWL. Median LC-excess of snowpack samples was close to rainfall (0 ‰), suggesting no significant snow sublimation in Vallon de Nant. The spring and stream samples showed negative median LC-excess values, indicating that recharged water has undergone some evaporation, which may vary over space and time. Out of all the springs, ICEC spring samples seemed to be less affected by evaporation (as shown by higher LC-excess value), suggesting that rainfall over the area upstream of this spring directly infiltrated into the ground. This is further supported by the presence of sparse vegetation in this part of the catchment.

### 3.5.5    $^{17}O$-excess

We observed that snow has a median value of $^{17}O$-excess of 91.3 per meg and is significantly higher than that of rainfall, 49.2 per meg. The difference between rainfall, snowpack and glacier observed for $\delta^2H$, $\delta^{17}O$ and $\delta^{18}O$ is also visible with $^{17}O$-excess, as opposed to d-excess. $^{17}O$-excess could potentially be useful to distinguish between rainfall, snowpack, and ice melt but secondary evaporative processes complicate a direct interpretation.

Given that the local and global reference lines for $^{17}O$ are very similar (see Section 2.3), it is tempting to interpret the spatial differences in $^{17}O$-excess values; the median values of all sampled water show a coherent picture, with subsurface and stream water having intermediate values between rainfall and snow samples and thus being a mix thereof. As for d-excess, we can however not draw any direct conclusions on mixing ratios since rainfall and snowfall undergo further evaporative processes during recharge. Furthermore, the temporal dynamic of $^{17}O$-excess in springs does not show additional information compared

to d-excess. Since use of $\delta^{17}O$ and $^{17}O$-excess for tracing of alpine hydrologic processes is still relatively new, our discovery that they offer limited immediate added value is useful for future studies.

## 4 Discussion

### 4.1 Comparison of similar tracer studies in Alpine regions

Electrical conductivity of all samples is higher compared to previous such studies in Alpine environments (Cano-Paoli et al., 2019). This suggests that our site has comparatively more exchange with rocks and sediments. The temporal evolution pattern of EC that we see is typical (Penna et al., 2014; Cano-Paoli et al., 2019), as is the time lag between seasonal cycles in EC and streamflow (Cano-Paoli et al., 2019). On an event-scale basis, this lag between streamflow and EC was explained by the delay between the transmission of pressure waves (leading to discharge increase) and the actual arrival of newly recharged water
(Chiaudani et al., 2019).

The median $\delta^{18}O$ value of all our streamflow samples (-12.7 ‰) is slightly lower than values observed for the Rhone in Porte de Scex (Schurch et al., 2003), of which Vallon de Nant is a headwater catchment (albeit one with relatively low elevation compared to other headwater catchments of the Rhone). The slightly lower values can be explained by the proportion of snow to rain in the specific headwater region of this study relative to other regions. The isotopic lapse rate that we observe in
precipitation is half that estimated for Switzerland based on data from the Global Network of Isotopes in Precipitation (GNIP) between 1966 and 2014, that is -1.9 ‰/100/m for $\delta^2H$ and -0.27 ‰/100/m for $\delta^{18}O$ (Beria et al., 2018).

Our finding that streamflow is composed of previously stored ground water rather than of recent mid-winter snowmelt from hydrologically proximate areas (e.g. in the floodplain or the riparian area) based on the lack of change of $\delta^{18}O$ (and $\delta^2H$) during E directly contradicts assumptions of some snowmelt-runoff models (Schaefli et al., 2014).

The deuterium excess (d-excess) is in the range of published values for rainfall in the Swiss Alps (Leuenberger and Ranjan, 2021) as is our higher value from snowpack (Grimsel, Schotterer et al., 2004). Higher d-excess in snowpack in these regions is related to the presence of a secondary source of winter precipitation, which comes from the Mediterranean Sea (Froehlich et al., 2002) versus the dominant precipitation source being the Atlantic Ocean during majority of the year (Sodemann and Zubler, 2009). Secondary evaporative processes within the snowpack shift their isotopic ratios away from the LMWL, along
the local evaporation line, causing a decrease in d-excess values (Beria et al., 2018).

Our computed $^{17}O$-excess values of rainfall (Figure 4 G) are much higher than the few published values in Switzerland, which range from 6.5 per meg (Leuenberger and Ranjan, 2021) to 18 per meg (Affolter et al., 2015) for low and high elevation locations. There are no published values for snowfall or snowpack for Switzerland, but values between 17 and 62 per meg for freshly precipitated snow on Mount Zugspitze (German Alps, 2962 masl) are found in the work of Surma et al. (2021).
Variations in $^{17}O$-excess have been found to be affected my local meteorological factors such as precipitation formation or as tracers of the evaporative conditions at the moisture source season-by-season (e.g., precipitation in summer but not in other seasons) in other regions of other continents (Midwestern North America, Tian et al., 2018). It is possible that the variations

we observe are limited due to the scale of our site. The difference in values between our results and those from other studies are unlikely to be related to the analytical approach used, as the normalization of the data was done using standards cross-calibrated in several laboratories also using a gas-source mass spectrometer approach. Similarly, we adopted measurement strategies to limit likely memory effects, notable for measurements of $\delta^{17}O$ values and $^{17}O$-excess (Vallet-Coulomb et al., 2021). Thin snowpack has been found elsewhere to result in negative temperatures in the soil, as we measured during the 2016-2017 winter, which was quite dry, at 1530 m and 2640 masl (Bender et al., 2020).

## 4.2 Dominant streamflow generation processes

We conceptualize the dominant hydrologic processes during different times of the year in Figure 8. During the winter baseflow period (Figure 8a), large parts of the catchment are covered with snow. Streamflow is very low as streams are mostly supplied by groundwater, with episodic melt events occurring in lower parts of the catchment. The isotopic ratios of these melt events are more depleted in heavier isotopes compared to groundwater. During the early melt period (Figure 8b), the snowpack at lower elevations and close to the stream network starts releasing water to the subsurface, without melting completely. Streamflow during this period is predominantly driven by groundwater; the contributions from snowmelt and potentially rainfall at lower elevations do not lead to a change of the isotopic values of streamflow compared to the winter recession. Subsequently, snowmelt intensifies across the catchment (Figure 8c), significantly increasing the saturated subsurface areas and catchment connectivity. Streamflow during this period is dominated by snowmelt (which largely flushed the subsurface water storage), as is common during the freshet (Frisbee et al., 2011), resulting in very depleted isotopic ratios in stream water. Once the snowpack melts out (Figure 8d), streamflow is driven by episodic rainfall events. We will now discuss the dominant recharge processes during different parts of the year.

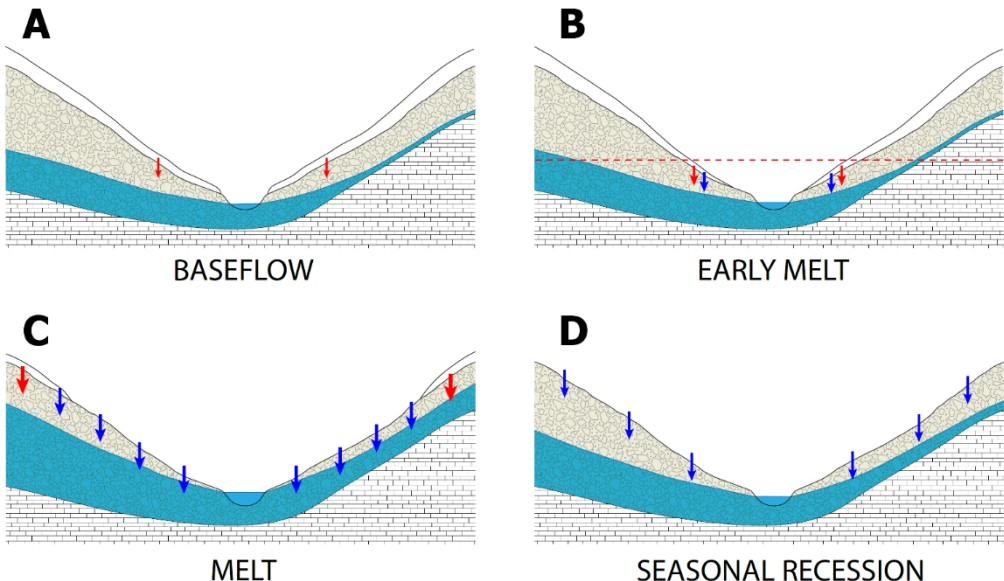

Figure 8. Conceptual figure illustrating the four streamflow generation periods: Baseflow (A), Early melt (B), Melt (C), and seasonal Recession (D). Red arrows represent water depleted in heavier isotopes (for example, snow melt), and blue arrows represents water enriched in heavier isotopes (i.e., rain). Magnitude of flux (size of arrows), location of flux, distribution of saturated area (blue), and snow cover (white) vary between the four seasons. From left to right, top to bottom: Baseflow shows a minimal contribution of snow melt to subsurface and surface even during winter; during Early melt the snow recedes at the lower elevations first (red dashed line) and contributes to streamflow; the Melt period is characterized by retreating snow cover, fluxes dominated by rain input, and small snow contributions at high elevation; Finally seasonal Recession consists only of rain inputs. Asymmetric hillslopes with differing reservoir sizes and hydraulic conductivities may produce different responses during the different phases.

### 4.2.1    During winter

During winter, streamflow follows a long recession curve, with almost constant baseflow between January and March. Such a constant winter baseflow can result either from a prolonged emptying of a groundwater store (Cochand et al., 2019), as assumed in many rainfall-runoff models (e.g. Muelchi et al., 2021; Staudinger et al., 2017) or from an interplay of groundwater recession and small amounts of snowmelt occurring at the snow – ground interface in absence of soil freezing (Schaefli et al., 2014). Our streamflow isotope measures suggest that streamflow is indeed resulting from a long recession and thus indicative of sufficient groundwater storage to sustain winter baseflow. However, we observed diverging trends in isotopic ratios in two springs: The lower elevation spring located near the catchment outlet (AUBG) showed depletion in heavier isotopes during winter, whereas the spring located in the higher part of the catchment (BRDG) showed enrichment in heavier isotopes (Figure 5). This suggests that there are contrasting processes at play at different elevations in Vallon de Nant during winter, which has also been reported previously in snow influenced catchments, such as in the Colorado River basin (Carroll et al., 2019). As snowmelt is more depleted in heavier isotopes compared to average annual groundwater recharge (Figure 4), the lower

elevation spring (AUBG) is likely influenced by winter snowmelt, as previously observed in other Alpine catchments (Rücker et al., 2019). However, the higher elevation spring (BRDG) is largely influenced by underlying groundwater storage, which is more enriched in the heavy isotopes compared to snowmelt (Figure 4). This is further supported by increasing trends in EC values at BRDG over the course of winters (Figure 5d), indicating the prominence of subsurface flow. The lower elevation AUBG spring shows slightly decreasing EC values over winters, confirming the presence of winter snowmelt. Thus, we conclude that in Vallon de Nant, winter baseflow at lower elevations is the combined result of a long seasonal recession and of winter snowmelt (Figure 8a). At higher elevations, winter baseflow can be largely explained by the underlying groundwater, and hence the longer seasonal recession. It is however unclear if snowmelt at lower elevations was due to atmospheric heat exchange or ground heat exchange, a question that could be interesting for future research.

### 4.2.2    During early spring snow melt

The early melt period is rarely discussed in the literature (for a model-based example, see the work of He et al., 2015), despite its importance in terms of water supplied to the catchment and prevalence across Alpine regions, and the streamflow during this period remains challenging to model (see Figure 9 in Brauchli et al., 2017; or Figure 3 in Thornton et al., 2022). In Vallon de Nant, the start of the early melt period coincides with thinning or local disappearance of the snowpack from lower elevation sites, as can be clearly seen in soil temperature measurements at lower elevations (Figure 3) (and as depicted in our sketch at Figure 8b). This suggests that streamflow rise during the early melt might be linked to snowpack melting at lower elevations. During this period, snow cover is still abundant at higher elevations as can be seen in soil temperature measurements at higher elevations (Figure 3). It is possible that snowmelt might be happening within the existing snowpack at higher elevations, but then getting retained within the snowpack (Gerdel, 1945) or be locally stored in the subsurface, and thus not leading to a strong streamflow response. An in-depth analysis is beyond the scope of this article but gives an interesting opportunity for future studies.

Interestingly, streamflow increases swiftly at beginning of E1 period, whereas EC and isotopic values ($\delta^2$H, $\delta^{17}$O and $\delta^{18}$O) show a lagged response (Figure 6a-d), suggesting that older water stored within the subsurface (with higher EC and more enriched in heavier isotopes) are first to be exfiltrated into the stream (Mcdonnell et al., 2010) at the onset of early melt. In other words, the streamflow reaction during this period is not resulting from localized snowpack outflow directly to the stream but from snowmelt transiting through the subsurface. Accordingly, during this period, streamflow rise is most likely limited by both limited snowpack outflow and temporary retention of water in the subsurface.

### 4.2.3    During melt periods

During the melt season, isotopic compositions of the springs converge to a common value (see M1 period in Figure 5) (around -93.5 ‰ for $\delta^2$H, -6.8 ‰ for $\delta^{17}$O and -13.0 ‰ for $\delta^{18}$O), suggesting that the entire subsurface gets saturated (Figure 8c) and flushed with snowmelt. This convergence to similar values during the melt is a priori surprising because: i) we have shown an isotopic lapse rate for precipitation of 0.84 ‰/(100m) for $\delta^2$H and of 0.128 ‰/(100m) for $\delta^{18}$O; and ii) the annual median

values of the isotopic values for different springs decrease with elevation (Figure 4a-c). Accordingly, the convergence of isotopic compositions of the springs during melt to a common value strongly hints towards melt water coming from a similar elevation range or rather towards melt water being sampled from different elevations in a similar way. This has also been seen in previous studies in mountainous catchments (Feng et al., 2022; Penna et al., 2017).

The higher EC values in the stream at certain instances of the melt period compared to springs (Figure 5, Figure 6) are unexpected and might suggest that there is a significant amount of subsurface water reaching the stream that has higher EC values than all sampled springs. This result however underlines the importance of subsurface flow paths to stream recharge during the melt periods. The positive temperature anomalies (during summer rainfall events) observed during M2 (ROCK) show the existence of fast surface flow paths but are not enough to explain the high EC values during this period.

### 4.2.4    During the seasonal recession

The EC values in all springs increase during the recession period, clearly suggesting that they are all fed by subsurface water during this period of the year. The isotopic values of the springs diverge however after the melt period, giving insights into the underlying reservoirs that are feeding them, and their relative permeabilities and outflow rates. A smaller increase in δ-values indicates a larger subsurface reservoir or slower flow rates/permeabilities (e.g., BRDG), where a larger increase is associated with a smaller reservoir size or higher flow rates / permeabilities (e.g., AUBG). This difference between BRDG and AUBG springs suggest existence of multiple subsurface reservoirs in Vallon de Nant, which has previously been observed in other high elevation landscapes (Dwivedi et al., 2019; Mosquera et al., 2016).

The isotopic values in streamflow gradually increase over the seasonal recession period towards those of groundwater, suggesting that groundwater becomes the dominant contributor to streamflow generation during this part of the year (Figure 6). This can also be seen in increasing EC values of streamflow in the R2 period, suggesting prevalence of deeper flow paths over shallower flow paths. However, during large rainfall events, there are sometimes sudden increases in stream isotopic ratio, suggesting that rainfall can become a significant contributor to streamflow generation during intense rain storms. The role of event-based precipitation on streamflow generation has long been studied (Mcdonnell, 1990; Kirchner, 2003; Kienzler and Naef, 2008) with most of the studies concluding that rainfall helps mobilize older water stored within the catchment and that hence, the isotopic ratio in stream water often largely resembles subsurface storage (see also the metanalysis of Barthold and Woods, 2015). Our measurements clearly suggest that this so-called "old water paradox" does not hold for some rainfall-generated streamflow responses during the recession period (Mcdonnell, 1990; Mcdonnell et al., 2010). What processes might explain the fast contribution of recent rainfall to streamwater during the recession in the Vallon de Nant remains to be studied in detail, based namely also on our results on rainfall patterns and their relation to the stream network (Michelon et al., 2021b). This analysis would however require additional analyses of shallow groundwater dynamics and the evolution of hillslope connectivity during rainfall events.

### 4.3 Water temperature reveals seasonal hydrologic connectivity patterns

Temperature measurements can provide interesting insights into the seasonal evolution of hydrologic connectivity (Miralha et al., 2023), in particular concerning the interaction between the alluvial plains and the hillslopes. During melt period M2, temperature in springs BRDG and PZ3 are correlated with streamflow variations, with BRDG spring showing positive anomaly and PZ3 spring showing negative anomaly. The positive anomaly at BRDG suggests that snowmelt water traversed the landscape, got heated by direct solar radiation and then infiltrated into the spring. This argument is also supported geologically,

as BRDG is recharged by snowmelt from nearby riparian areas and steep slopes facing west that are directly exposed to the sun (see Figure 1). In contrast, the negative anomaly at PZ3 suggests that snowmelt directly infiltrates during the melt period, also seen in the nearby areas that become snow free quickly (200m distance from soil temperature sensor at 1530 masl). This clearly differentiates the local snowmelt processes (in the order of tens of meters) from the more regional snowmelt patterns (in the order of few hundred meters).

Additionally, temperature measurement provides insight into groundwater connectivity. Water temperatures are usually influenced by ground temperature, but the high hydraulic conductivity in areas surrounding PZ3 does not allow enough time for the water temperature to reach equilibrium. This temporary (6 weeks) and local snowmelt input is superimposed on a longer scale pattern that leads to 74 days of lag between PZ3 and air temperature. This suggests that subsurface in Vallon de Nant gets fully saturated and well-connected during melt periods, and less connected during later part of the year (also depicted in

Figure 8c). This was also highlighted by stable water isotope measurements discussed in preceding sections, where the entire catchment becomes well connected during melt periods (Figures 5, 6).

The lags in water temperature can indicate the reactivity of the subsurface flow. The PZ1 spring (470 m to the north) reacts in a different way, with a 58-day lag indicating a shallower flow path, but without temperature anomaly during the melt period. Short-term temperature anomalies (positive during the summer, negative during the winter) associated with rainfall events

suggest local incursions of surface water, which is in contradiction with the absence of temperature anomalies during the melting period. One possible explanation is that the subsurface volume feeding this spring is small enough (with water levels between 0.8 and 2.4 m below the surface, see Supplementary Material, Figure S8) during R2 and B2 to react quickly to local surface inputs, while during M2 period volume increases significantly (with water level between 0.1 and 1.0 m below the surface) to not show short term reactions to melt water input. This highlights a very dynamic subsurface system.

Average temperature of subsurface water provides insight regarding the origin of the water. The average temperature difference between PZ1 and PZ3 (mean 6.3°C and 4.8°C over the year) can most likely be explained by their respective sub catchments: PZ1 (left bank) collects water from the grassy slopes of the west side of the valley (facing east), while through PZ3 (right bank) flows water from the south (facing north), with more shaded areas and snowpack remaining later in the year.

At the end of B2, temperature at the 4 springs converge to 4.3°C and if we limit ourselves only to this variable, we could think

that this is pointing toward a common aquifer feeding them during baseflow. The shift of the PZ1 and PZ3 temperatures (+0.4 °C and -0.5 °C) at the end of baseflow could be explained by an unconfirmed contamination issue, specifically of air into the

piezometers. The fact that isotope composition of the water in the streamflow during B2 is close to the median value of all sampled water sources suggests that our spatial sampling was good enough to represent the main water sources during baseflow.

The EC measurements clearly suggest that the subsurface flow path distributions are very similar in the upper part of the catchment (HyS2) and in the lower part of the catchment (HyS1). This is further supported by the fact that the isotopic lapse rate measured in rain water does not translate into streamflow.

The isotopic composition of GRAS is quite different from the other sources (mean values of -85.3 ‰ for $\delta^2$H, -6.3 ‰ for $\delta^{17}$O and -12.0 ‰ for $\delta^{18}$O). The absence of a temperature anomaly during the melt period suggests a large and well-mixed source

of water. The high thermal connectivity with the surface could then be explained by a shallow flow path over a certain distance before the water exits at the source. However, we still cannot explain why the temperature signal shows a variation induced by rainfall, whereas there is no variation due to snowmelt input.

## 4.4    Transferable insights

### 4.4.1    Temperature of water due to origin and shallow soil interaction

Although temperature is not a conservative tracer, temperature measurements of springs are useful to estimate flow path depth. We provide a quantitative tool in the Appendix 2. However, the underlying assumption of our method that heat transfer is driven by conduction might not always be verified (Kane et al., 2001), and anomalies between measured and modelled temperature (pure sinusoid) could be related to heat transport with subsurface water flows (i.e. to advection phenomena).

At shallow depths (10 cm), soil temperature is strongly influenced by air temperature, and our analysis of soil temperature at

different elevations shows that it can be used as a good proxy for detection of snow cover. Early melt starts when the soil temperature at low elevation (1240 masl) rises, showing that snow is melting near the catchment outlet. The soil temperature sensor, albeit not intended for this use, seems to be well positioned to detect the onset of early melt at lower elevations.

For the other soil temperature recordings at higher elevations, there is no direct link to streamflow dynamics. The time elapsed between snowcover disappearance at the soil temperature site at 1530 masl and the beginning of the melting period varies

significantly but is always positive (8 days in 2016, 3 days in 2017 and 51 days in 2018), underlying that intermediate elevations are actively contributing snowmelt water during the rise of the melt period. For the highest site, there can be some snow left at the beginning of the recession period.

Other studies have also observed the reactivity of soil temperature to snowpack depth. Bender et al. (2020) observed that in general ground temperature is more stable under thicker snowpacks and that high level of temperature variability in ground

temperature mainly occurs when the snowpack is absent, but they do notice that under thin snowpacks, ground temperature can fluctuate dramatically, though without heating, perhaps suggesting the presence of sub-snowpack moisture flow. In fact, they showed that vegetation combined with thin snow delayed the ground warming and discuss in detail the value of soil temperature recordings under snow. A larger number of soil temperature sensors would provide an interesting perspective to

identify more precisely the relative contributions of the different landscape units, elevations, and terrain aspects. This could
be particularly promising in combination with satellite images for snow cover mapping.

### 4.4.2     Isotopic composition of springs and stream water

Stable isotope composition of water is particularly promising to track the co-existence of seasonal baseflow and winter melt within springs and shallow groundwater. However, this requires a year-round time series to observe where and when isotopic ratios in groundwater becomes enriched or depleted. This year-round monitoring is particularly important as many subsurface
signals are likely to see a "reset", or return to baseflow, during the main melt period, as highlighted in our work here.

The range of isotopic composition for each water source informs on the relative snowmelt proportions from their respective sub-catchments. Without evidence of a strong isotopic lapse rate in snowfall, the differences measured can be explained by the variation of snowfall amounts with elevation.

The relative proximity of some water sources monitored in this study underlines that spatial proximity does not necessary
imply similar behaviors (in terms of temperature or isotopic composition), as we see noticeable differences between the sources due to the different characteristics of their sub catchments (i.e., flow path depth, hydraulic conductivity, slope, aspect).

LC-excess values might reveal some additional insights in future work, in combination with future analyses of soil water isotopes (to give insights into evaporation effects).

The d-excess and $^{17}$O-excess are typically used to investigate the large-scale hydrological cycle and oceanic moisture sources
(Nyamgerel et al., 2021). Both d-excess and $^{17}$O-excess respond to relative humidity during evaporative processes but $^{17}$O-excess may be less temperature sensitive (Surma et al., 2021; Bershaw et al., 2020) than d-excess and thus indicate compositions of evaporation and climatic conditions even when they would be invisible with d-excess (Risi et al., 2010). Our application is quite different than this typical, until now, application. At this stage, it is not clear either what the value of $^{17}$O-excess is for hydrological purposes and the question whether it conveys local scale information remains open. While so far not
reported for measurements of $\delta^{17}$O, it is known that under certain environmental conditions, the snowpack will experience melting, evaporation, and refreezing, forming a firn rather than snow. This may influence the $^{17}$O-excess measured for either fresh snow or the final snowpack (see, for example, Risi et al., 2010). Although, much laboratory time was devoted here to the measurement of $\delta^{17}$O and $^{17}$O-excess, we gained few insights or specific added value as we hoped for documenting the influence of local-scale snow dynamics, specifically the variation in space and time of accumulation, transport, storage, melt
and sublimation, on hydrological processes, except some unvalidated potential to distinguish glacier melt from snowmelt when combined with temperature measurements. Perhaps our measurements would have been more relevant if fresh snow had been sampled instead of the snowpack or if more relevant reference data were available. However, even if we cannot draw any concrete conclusions at this moment, we hope that the full value of the $\delta^{17}$O data set presented here will be investigated and prove independently valuable in the future.


### 4.4.3 The added value of EC

EC allows qualitative estimation of stream water sources and is very useful to distinguish periods where streamflow is dominated by snowmelt vs groundwater. During periods of high snowmelt, a drop in stream EC can be explained by dilution from meltwater (Chiaudani et al., 2019), which has shorter subsurface residence time vs older water stored within the subsurface, and accordingly a lower ionic content and EC (Cano-Paoli et al., 2019). However, the difficulty to characterize the different physical and geochemical properties of soils (influencing EC) do not allow an intercomparison of absolute EC values between the sources. However, variations at a given source may inform on the snowmelt input (low EC) or the flow path dynamic (old water pushed by water input). In catchments like Vallon de Nant that exhibit little elevational gradients in stream isotopic ratios, EC represents an extremely valuable tracer to segregate the relevance of snowmelt in streamwater generation.

## 5 Conclusion

This paper focuses on understanding the interplay of runoff generation processes in all four seasons in the high Alpine Vallon de Nant catchment using four tracers: water and soil temperature, EC, stable water isotopes. Furthermore, we discuss the value of these four tracers for hydrologic process investigation for comparable catchments. A future investigation in this location could render our conclusions more quantitative by using a modelling approach to explore our observations.

Streamflow generation in Vallon de Nant is the outcome of a complex temporal succession of surface and subsurface contributions that can be best understood by starting the analysis at the observed "reset" of the isotopic composition during the melt period, or freshet, which is when it returns to the baseline value. At this moment, the isotopic composition of the springs converge, the entire subsurface gets saturated and flushed with snowmelt that either comes from a similar elevation range or gets sampled from different elevations. Accordingly, interpretation of the isotopic dynamics becomes extremely complicated, and the value of year-round water isotope samples might be reduced to being a simple measure of the relative proportions of snowmelt compared to rainfall in the different sampled water sources.

The sampled EC values, in addition to isotopes, give further insights into subsurface exchanges in the Vallon de Nant catchment. Together, the isotopic composition and the EC values suggest that: i) subsurface flow plays a prominent role through all stages of snowmelt; and that ii) winter streamflow might be composed of both local winter snowmelt and groundwater contributions that are recharged on an annual basis. Temperature measurements in springs and soil across elevation gradients provide additional insights into flow path depth and highlights the effect of rain-on-snow events on soil temperature below the snowpack, though they are undetectable in discharge.

Based on our case-study specific conclusions, our take home messages for future work at other locations are:

- Understanding the dynamics of stable water isotopes in comparable high-elevation catchments requires sampling during the snow melt period in order to capture the moment that the values return to the baseline, which necessitates collecting samples year-round or at a minimum, starting very early in the melt season (which is often impossible responsibly at avalanche-prone locations).

- The complete return to the baseline, or reset, makes the interpretation of stable water isotopes samples from different surface and subsurface water sources particularly challenging and a combination with other tracers might be required for all similar studies. We recommend that EC monitoring be explored as a more direct indicator of water age and subsurface flow. Water temperature is recommended to add insights into how well different water stores are connected. Combined with EC, water temperature has a particular added value to disentangle shallow from deeper flow paths (as both can have high EC).

- Appropriate characterization of the variability of all tracers studied here (water and soil temperature, EC, stable water isotopes) requires sampling during the winter baseflow period, which is, again, a challenge at many places.

- Characterisation of different streamflow generation periods and processes can greatly benefit from continuous soil temperature measurements, which give information on presence and absence of an insulating snowpack and on potential disconnection of the subsurface in case of soil freezing. We recommend systematic soil temperature measurements in comparable hydrological process studies.

- Winter melt and runoff generation processes might be more present at high elevations than previously thought. These winter processes potentially condition the catchment-scale hydrologic response during the melt period and groundwater recharge at the annual scale. Future work on this topic should also revisit the concepts that correspond to this field-scale process in hydrologic models.

*Data availability*. Stable water isotopes and conductivity measures of each water sample used for this paper is available online https://doi.org/10.5281/zenodo.5940044 (Michelon et al., 2022). Access to other data is mentioned in the text.

*Author contributions*. AM and NC conceived the field study; AM, NC, HB, and JL collected and analyzed the field data; AM, NC and HB did all the lab work; all authors discussed and interpreted the data; AM wrote the first manuscript version, produced all computations and figures. BS and TV edited the first version of the manuscript, on which all authors gave comments. NC led the revision process and became corresponding author at this stage; all co-authors gave significant input on the revised version.

*Competing interests*. Author Bettina Schaefli was a member of the editorial board of the HESS journal, but otherwise, there are no other competing interests of which the authors are aware.

*Financial support*. This research has been funded by the Swiss National Science Foundation (SNSF; grant no. PP00P2_157611).

*Acknowledgement.* Thanks to all people who contributed to the field work for their precious help, namely Lionel Benoît, Tristan Brauchli, François Mettra, James Thornton, Inigo Irarrazaval Bustos, Tom Müller, Pascal Egli, Loïc Perez, Aurélien Ballu, Judith Eeckman, Mirjam Scheller, Marvin Lorff, Rokhaya Ba, Anham Salyani, Guillaume Mayoraz, Raphaël Nussbaumer, Emily Voytek, Micaela Faria, Michael Rowley, Guillaume Gavillet, Gelare Moradi, Gabriel Cotte and Moctar Démbélé. Thanks to Markus Randall and Loïc Perez for their help measuring electric conductivity in the lab.

**Appendix 1: Influence of air on the isotopic composition of a water sample ($\delta^2H$, $\delta^{17}O$ and $\delta^{18}O$) within a sealed container**

The purpose of this calculation is to estimate how the isotopic composition of a water sample locked up together with some air in a sealed container will be altered by the water vapor of the air. This configuration may happen i.e., with snow sampling as snow density ranging from 0.55 to 0.83 suggests that at least 17 % to 45 % of the volume in the container is ambient air

from the sampling site. To make these calculations we consider the conditions in which the samples will be analysed; we take the ambient temperature of 25.3 °C for which we know the isotopic fraction factor between vapor and liquid phases of water for $\delta^2H$, $\delta^{17}O$ and $\delta^{18}O$. At this temperature the samples are in a liquid phase, and in equilibrium with the air of their container. Following Mook (2008), the isotopic fractionation of water between two phases at the equilibrium is written as a reaction between the liquid $l$ and vapor $v$ phases of $H_2O$ as:

$$H_2O_l + H_2O^*_v \leftrightarrow H_2O^*_l + H_2O_g, \tag{1}$$

where * marks the heavy isotopic form of the molecule that may contain $^2H$, $^{17}O$ or $^{18}O$, and $\delta^*$ its isotopic composition in per mil. At a given temperature $T$, the isotopic fractionation factor of water between liquid and vapor $\propto_{l/v}$ is the equilibrium constant of the Equation 1:

$$\propto_{l/v}(T) = \frac{[H_2O^*]_l[H_2O]_v}{[H_2O]_l[H_2O^*]_v} = \frac{\delta^*_l/1000+1}{\delta^*_v/1000+1} \tag{2}$$

As we know: i) the amount of liquid water in the container and its initial isotopic composition; ii) the amount of ambient air captured in the container and its initial isotopic composition; and that we can deduce iii) the total amount of heavy isotopes in the total amount of water, we can solve the Equation 2 as a second order equation.

The calculations are made for two extreme amounts of air vapor saturation, namely air without any water vapor and air fully saturated with water vapor. For the last one we take the partial pressure of water at 25°C P=3169.9 Pa (Haynes et al., 2017):

The value of the fractionation factor of water $^2H$ and $^{18}O$ between 0 and 100°C are (Majoube, 1971):

$$ln\ ^2\propto_{l/v}(T) = 24.844.10^3/T^2 - 76.248/T - 52.612.10^{-3} \tag{3}$$

$$ln\ ^{18}\propto_{l/v}(T) = 1.137.10^3/T^2 - 0.4156/T - 2.0667.10^{-3} \tag{4}$$

From Equations 3 and 4 we compute $^2\propto_{l/v}(T = 25.3\ °C) = 1.0789$ and $^{18}\propto_{l/v}(T = 25.3\ °C) = 1.0135$.

For $^{17}O$ we will take the experimental values given by Barkan and Luz (2005) at 25.3 °C: $^{17}\propto_{l/v} = 1.00496 \pm 0.00002$.

For each stable water isotope, the values are calculated for 2 extreme sample isotopic composition from our database ($\delta^2H = -180$ ‰ and 5 ‰, $\delta^{17}O = -12$ ‰ and 0 ‰, $\delta^{18}O = -30$ ‰ and 5 ‰). The range of the isotopic composition of ambient air is based on records reported by Wei et al. (2019) for Rietholzbach, Switzerland (755 masl) from August to December 2011: the $\delta^2H$ air values range between -239.79 ‰ and -73.48 ‰, and $\delta^{18}O$ values range between -31.41 ‰ and -9.94 ‰. No reference value is available for $\delta^{17}O$, so a range between -30 and 0 ‰ has been chosen arbitrarily.

The Figure A1 shows the changes of the sample isotopic composition for $\delta^2H$, $\delta^{17}O$ and $\delta^{18}O$. These values have been completed for different amounts of air (ratios of sample volume over container volume).

The constant error for dry air corresponds to the case where the water vapor in air originates via evaporation of the water sample.

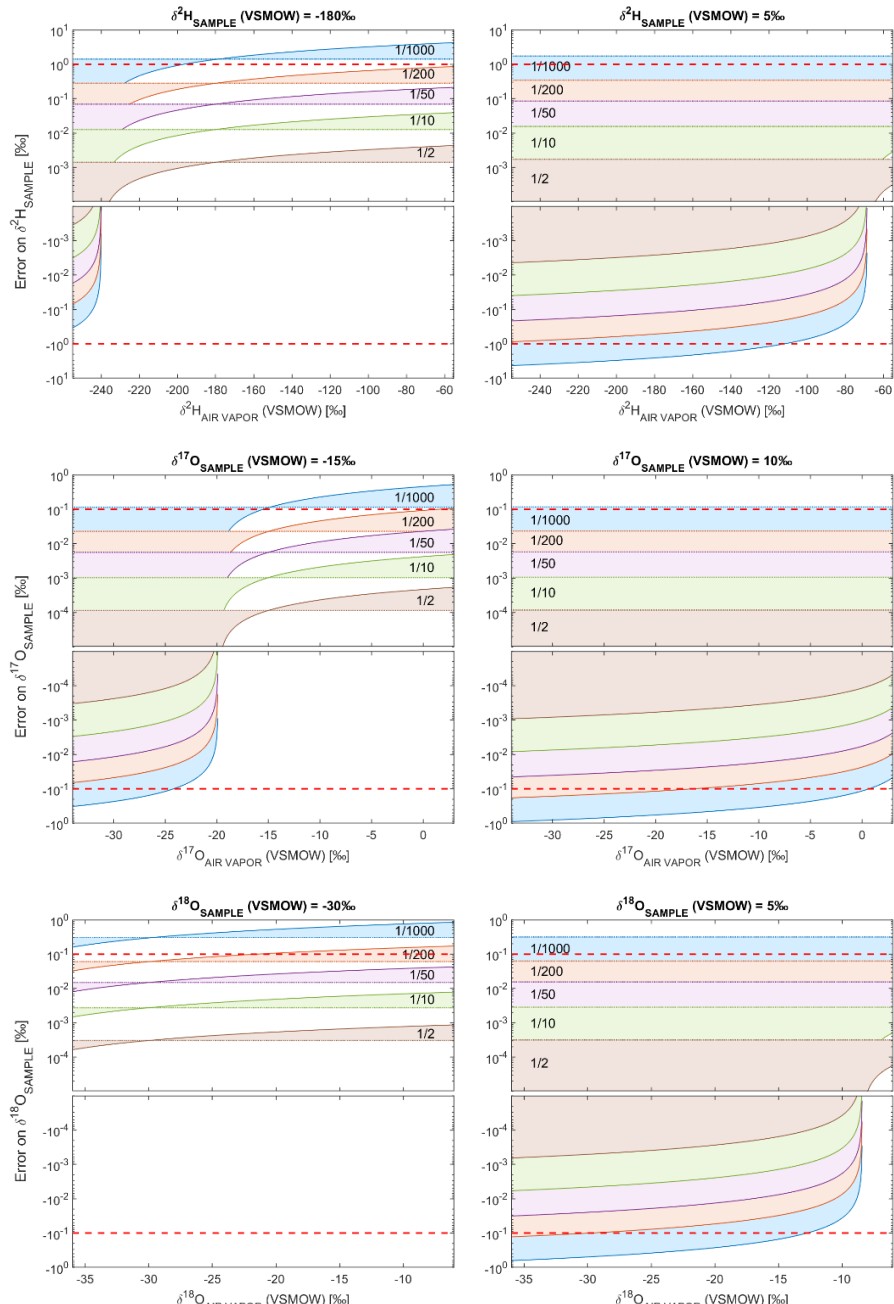

Figure A1: Changes in $\delta^2H$, $\delta^{17}O$ and $\delta^{18}O$ of a water sample depending on the initial isotopic composition of the water vapor of the air locked up with the sample. The results are completed for saturated air (continuous line) and dry air (dashed line), for ratios of sample volume over container volume from 1/2 to 1/1000. The red dashed line represents the detection limit of the measuring device.

**Appendix 2: Estimate of water flow depth based on a soil temperature model**

The estimate of the water flow depth is based on the soil temperature model presented in the work of Elias et al. (2004), assuming the water temperature measured at the spring/piezometer being equal to the soil temperature at the mean water flow

depth. The evolution with time $t$ of soil temperature $T$ at the surface (depth $z=0$) corresponds to air temperature, and is characterized by the mean air temperature $T_a$ and its amplitude $A$:

$$T(z=0, t) = T_a + A \sin(\omega t + \varphi), \tag{4}$$

with $\omega$ the radial frequency (in rad/s) and $\varphi$ a phase constant (in rad). The heat transfer into the soil is dampened by $D$, the dampening depth coefficient (in m) expressed as a function of K (in m²/s) the soil thermal diffusivity:

$$D = \sqrt{\frac{2K}{\omega}}, \tag{5}$$

giving the soil temperature at depth z:

$$T(z, t) = T_a + A \exp\left(-\frac{z}{D}\right) \sin\left(\omega t - \left(\frac{z}{D}\right) + \varphi\right), \tag{6}$$

The lag time $L$ between air temperature and soil temperature at a given depth z is then:

$$L(z) = \frac{z}{\omega D}. \tag{7}$$

The depth is approached using the *fminsearch* function in MatLab, reducing the error between the observed lag time and the modelized lag time. Although the thermal diffusivity of soil is influenced by: i) water volumetric content; ii) volume fraction of solids; and iii) air-filled porosity (Ochsner et al., 2001), we retain for this computation a unique value of thermal diffusivity of soil for all the points, using the typical value of 5.56.10⁻⁷ m²/s (Elias et al., 2004). The sinusoidal air temperature is based on time series from a grided product (1 x 1 km grid) from MeteoSuisse (Schaefli, 2021). The results are presented in Table 2

and Figure A2.

Table 2. Characteristics of the sinusoidal air and water temperatures used for the soil temperature model, and characteristics of the soil temperature at the estimate depth corresponding to the water temperature.

| Water sources | Measured air T [°C] | | Measured water T [°C] | | Air/Water lag time [d] | Modelized soil T [°C] | | Modelized Soil depth [m] |
|---|---|---|---|---|---|---|---|---|
| | *mean* | *amplitude* | *mean* | *amplitude* | | *mean* | *amplitude* | |
| PZ1 | 5.8 | 10 | 6.3 | 3.8 | 79 | 6 | 5.1 | 3.2 |
| PZ3 | 5.8 | 10 | 4.8 | 3.7 | 105 | 4.5 | 3.3 | 4.3 |
| GRAS | 6.3 | 10 | 5.5 | 3 | 41 | 5.3 | 9.9 | 1.7 |
| ROCK | 6.3 | 10 | 5.4 | 2.5 | 39 | 5.2 | 10.2 | 1.6 |
| BRDG | 5.8 | 10 | 4.7 | 0.9 | 6 | 5 | 18 | 0.2 |
| ICEC | 5.8 | 10 | 4.3 | 0.4 | 133 | 4.2 | 2 | 5.4 |

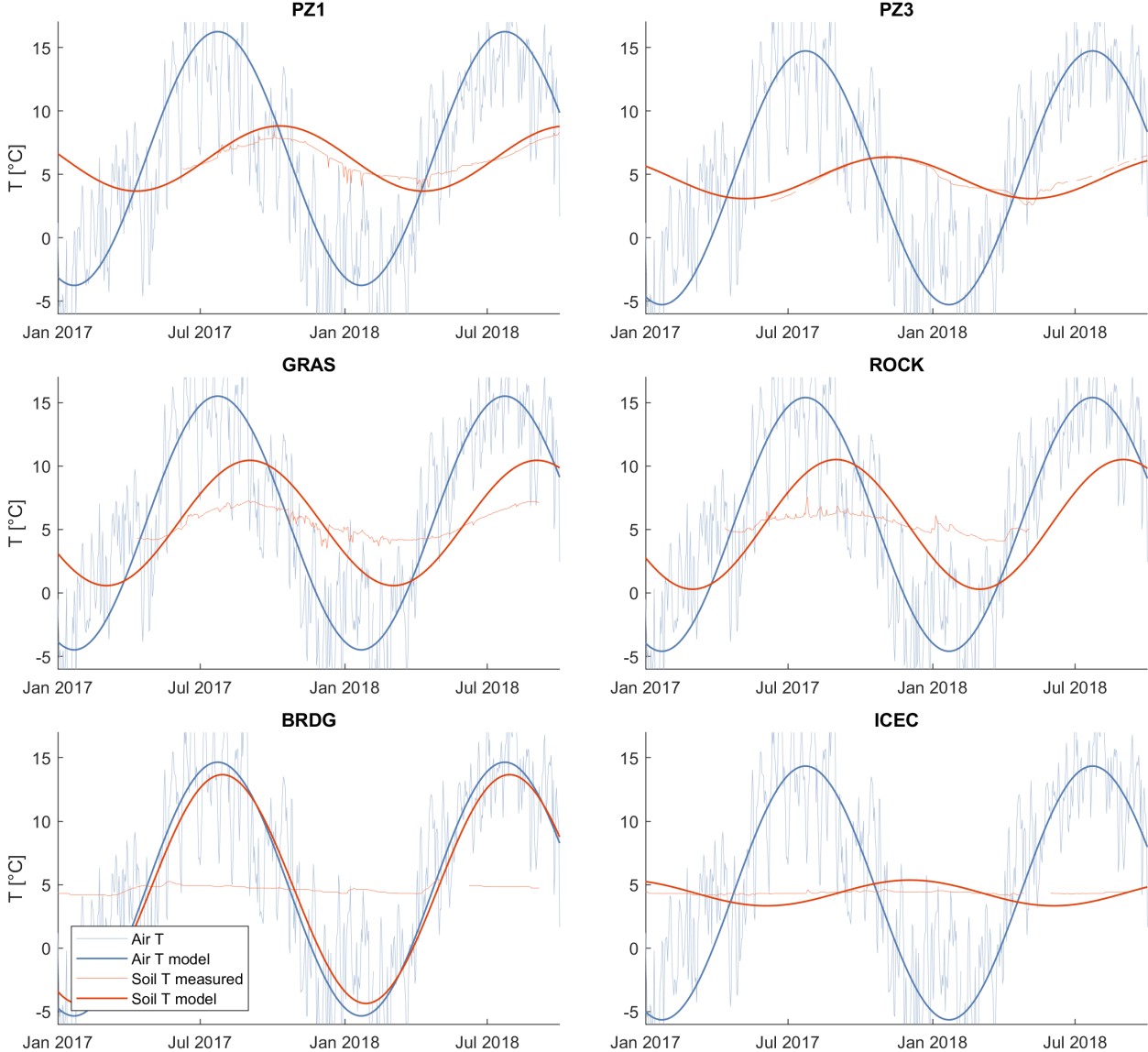


Figure A2: Measured and modeled air and soil temperature for 2 piezometers (PZ1 and PZ3) and 4 springs (GRAS, ROCK, BRDG and ICEC).

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
