# Peer review of "Hydrodynamics of a high Alpine catchment characterized by four natural tracers"

_Hydrology and Earth System Sciences, 2022_

## Author Comment (AC1)

**Supplement to reply to reviewer regarding hess-2022-48 Studying the dynamics of a high alpine catchment based on multiple natural tracers**

Anthony Michelon[1], Natalie Ceperley[2], Harsh Beria[3], Joshua Larsen[4,5], Torsten Vennemann[1], Bettina Schaefli[1,2]

[1]Institute of Earth Surface Dynamic (IDYST), Faculty of Geosciences and Environment (FGSE), University of Lausanne, Lausanne, Switzerland
[2]Institute of Geography (GIUB) and Oeschger Center of Climate Change Research (OCCR), University of Bern, Bern, Switzerland. Bettina Schaefli: Institute of Geography
[3]Department of Environmental Systems Science, ETH Zurich, Zurich, Switzerland
[4]School of Geography, Earth and Environmental Sciences, University of Birmingham, Birmingham, UK
[5]Birmingham Institute for Forest Research (BIFOR), University of Birmingham, Birmingham, UK

*Correspondence to*: Bettina Schaefli (bettina.schaefli@giub.unibe.ch)

Supporting figures and tables for replies to reviewer 1 submitted on Monday October 3, 2022. File contains 2 figures and 1 table. Figure 1 is the basis for a conceptual figure. Figure 2 is an illustration of the seven-day moving average. And the Table contains characteristic information regarding each of the defined flow periods.

[Figure]

*Figure 1. The basis for a conceptual figure illustrating four hydrologic seasons: baseflow (upper left), early melt (upper right), melt(lower left), and seasonal recession (lower right).*

[Figure]

*Figure 2. Illustration regarding the effect of a seven-day moving average on discharge measurements.*

| Period | Start Date | End Date | $Q_{m1}$ [m³/s] | $Q_{m7}$ [m³/s] | $Q_{r7}$ [m³/s] | HH:MM | $Q_b$ [m³/s] | $Q_f$ [%] | *dqdt* [m³/s/d] |
|--------|-----------|----------|-----|-----|-----|-------|-----|-----|------|
| B0 | 01.Jan.16 | 31.Mar.16 | 0.59 | 0.75 | 0.02 | 09:08 | 0.64 | 15 | 0.01 |
| E0 | 31.Mar.16 | 06.May.16 | 2.44 | 2.56 | 0.11 | 22:47 | 2.20 | 58 | 0.10 |
| M0 | 06.May.16 | 24.Jun.16 | 9.72 | 10.08 | 0.37 | 15:43 | 8.94 | 93 | 0.11 |
| R1 | 24.Jun.16 | 27.Sep.16 | 3.87 | 4.62 | 0.15 | 06:57 | 3.57 | 74 | -0.12 |
| B1 | 27.Sep.16 | 18.Mar.17 | 1.05 | 1.01 | 0.02 | 20:59 | 0.89 | 27 | -0.01 |
| E1 | 18.Mar.17 | 06.May.17 | 2.23 | 2.23 | 0.06 | 00:07 | 2.10 | 52 | 0.02 |
| M1 | 06.May.17 | 02.Jun.17 | 6.84 | 7.26 | 0.33 | 18:20 | 5.91 | 84 | 0.31 |
| R2 | 02.Jun.17 | 27.Sep.17 | 4.04 | 4.22 | 0.15 | 13:45 | 3.10 | 70 | -0.08 |
| B2 | 27.Sep.17 | 08.Apr.18 | 1.01 | 0.99 | 0.02 | 04:06 | 0.94 | 26 | -0.01 |
| M2 | 08.Apr.18 | 31.May.18 | 7.85 | 8.32 | 0.31 | 12:28 | 7.55 | 87 | 0.19 |
| R3 | 31.May.18 | 01.Jan.19 | 4.59 | 4.64 | 0.09 | 13:20 | 4.47 | 74 | -0.06 |

*Table 1. Table showing characteristic measurements for each period, corresponding start and end dates. Qm1 is the median of 1 minute data. $Q_{m7}$ is the median of daily values of average 7-day moving average data. $Q_{r7}$ is the range of 7 day moving average data possible in one day. HH:MM is the median time of peak flow in a day. $Q_b$ is the median baseflow, calculated as the lowest of the 7-day moving average flow in a ten-day window. $Q_f$ is the median of daily flow quantiles. dqdt is the change in flow according to a 7-day window around 1 day*

---

## Author Comment (AC2)

**Supplement to reply to reviewer regarding hess-2022-48 Studying the dynamics of a high alpine catchment based on multiple natural tracers**

Anthony Michelon[1], Natalie Ceperley[2], Harsh Beria[3], Joshua Larsen[4,5], Torsten Vennemann[1], Bettina Schaefli[1,2]

[1]Institute of Earth Surface Dynamic (IDYST), Faculty of Geosciences and Environment (FGSE), University of Lausanne, Lausanne, Switzerland
[2]Institute of Geography (GIUB) and Oeschger Center of Climate Change Research (OCCR), University of Bern, Bern, Switzerland. Bettina Schaefli: Institute of Geography
[3]Department of Environmental Systems Science, ETH Zurich, Zurich, Switzerland
[4]School of Geography, Earth and Environmental Sciences, University of Birmingham, Birmingham, UK
[5]Birmingham Institute for Forest Research (BIFOR), University of Birmingham, Birmingham, UK

*Correspondence to*: Bettina Schaefli (bettina.schaefli@giub.unibe.ch)

Supporting figure for replies to reviewer 2 submitted on Monday October 3, 2022. File contains figure 1, which is the basis for a conceptual figure.

[Figure]

*Figure 1. The basis for a conceptual figure illustrating four hydrologic seasons: baseflow (upper left), early melt (upper right), melt (lower left), and seasonal recession (lower right).*

---

## Author Response (AR1)

Dear Editor, Dear Reviewers

We would like to thank the reviewers for their detailed comments and the editor for handling our paper and giving additional advice. Below, we give detailed responses regarding how we modified the manuscript, and we include the track-change version of our manuscript. Our responses are introduced by double carrots >> after each reviewer comment.

Thanks to the reviewers' comments, the quality of the manuscript substantially increased, and we believe that it is now ready for publication.

On behalf of all authors,

Natalie Ceperley

**Reviewer 1**

We are very thankful for the time and feedback you gave us. Please find below a point-by-point response to your specific comments. In addition to general clarifications, we have 1) proposed a conceptual figure; 2) provided an additional figure and discussion on the purpose and effect of the moving average on the discharge time series; 3) explored how the streamflow periods B, E, and M relate to the stream flow percentiles. We have carefully revised the manuscript and incorporated your minor comments and technical corrections.

**Abstract**

The abstract is too generic and vague. I suggest rewriting it and make it more process-oriented so the reader can better appreciate what are the main findings and contribution of this work in terms of understanding the hydrological functioning of this catchment.

>> We have reworked the abstract to focus on the process-oriented process and highlight the main findings.

**Introduction**

1) The first part of the Introduction works well but what is missing is a clear statement about the main gaps in the current literature and the main research gaps that this work aims at addressing. Reporting that "this work attempts to quantify dominant drivers…" is ok for a general objective but this is not clearly link to a current lack of knowledge that this work could contribute to fill in. Please, rewrite lines 51-63.

>> We have rewritten lines 51-63 with a clearer statement:

> "To help fill some of the important knowledge gaps on elevational and seasonal drivers, this work uses high frequency tracer sampling to quantify drivers of the hydrologic response of a high elevation catchment throughout all streamflow periods, ranging from winter low flow, to different stages of the melt season and the autumn recession by compiling and complementing observational data from the intensively studied Vallon de Nant catchment in the Swiss Alps (Benoit et al., 2018; Giaccone et al., 2019; Ceperley et al., 2020; Mächler et al., 2021; Michelon et al., 2021b; Thornton et al., 2021a; Beria et al., 2020b; Antoniazza et al., 2022).

> The overall objective of this work is to examine dominant hydrological processes and associated flow paths in Vallon de Nant during different periods of the year through the lenses of four tracers: soil temperature, water temperature, electrical conductivity, and stable isotopes of water. The first tracer used in this study is the stable isotope composition of water, a natural tracer that has been extensively used to characterize snow hydrological processes (e.g. Beria et al., 2018), and is particularly useful to examine the interplay between

different water compartments (rainfall, snowpack, springs, groundwater), recharge and evaporation processes (e.g. Sprenger et al., 2016). Electrical conductivity measurements as an additional tracer provides information on subsurface flow paths and water residence times in the subsurface (Cano-Paoli et al., 2019). Water temperature measurements can be used to quantify connectivity between water sources and the atmosphere (Constantz, 2008). And lastly, soil temperature measurements are used to identify periods of thermal insulation from the seasonal snow cover (Trask et al., 2020)."

2) Similar, the specific objectives (64-70) are all lumped together, and this does not make justice to the worked carried out. I strongly suggest restating them as specific questions that can be measurable and are clearly linked to the different sub-sections of the results.

>> We have rewritten lines 64-70 with a more specific question:

"Using these tracers, we explore the origin of winter streamflow (the alteration and mixing of subsurface storage and from localized snow melt) (Floriancic et al., 2018; Hayashi, 2020), the dominant runoff processes that drive streamflow generation during early compared to late snow melt phase (Brauchli et al., 2017) and during the seasonal recession, the streamflow generated by shallow groundwater in the hillslopes and of alluvial or talus groundwater systems throughout the year (Hayashi, 2020). These explorations provide key transferable insights into the value of these four tracers for hydrologic process investigation that are relevant for comparable catchments."

Discussion

3) The weakest part of this manuscript is the Discussion:

A) it's highly fragmented in small subsections that, instead of making the structure clearer, produce a confusing picture.

>> The discussion mirrors our specific questions exactly, but to make that clearer, we have redefined the structure according to the following outline, with the 3 sections of dominant runoff processes grouped together as subsections:

- 4.1 Comparison with other alpine studies
- 4.2 Dominant streamflow generation processes
  - 4.2.1 During winter
  - 4.2.2 During early spring snow melt
  - 4.2.3 During melt periods
  - 4.2.4 During the seasonal recession
- 4.3 Water temperature reveals interplay of shallow and talus groundwater in hillslope
- 4.4 Transferable insights
  - 4.4.1 Temperature of water origin and shallow groundwater
  - 4.4.2 Isotopic composition of springs and stream water
  - 4.4.3 The added value of EC
  - 4.4.4 The uncertain value of $\delta^{17}O$ and $^{17}O$-excess

B) The results are discussed "internally" and there are almost no references to other works. This make it confusing to the readers to understand what is different and new compared to previous knowledge, and what is the contribution of this research.

>> Thank you for your point of view and pointers of key references, we have developed the discussion further along these lines, citing also other references to similar, published work.

C) As a results, the novelty, and the original contribution of this paper (which, in my opinion, exists) is hidden and not stressed at all, making it, unfairly, a general case study.

>> As we revised this manuscript carefully, we have emphasized the unique contribution of this study to hopefully move it out of the general case study category.

D) As a results of the confusing structure of this Section (point 1), the most important finding of this work is not very clear to the reader as it could be, i.e., how this catchment responds to precipitation and snowmelt inputs! Time series and reported ranges of values are of course important but I suggest adding a sort of conceptual graph showing the main runoff generation processes and the general behavior of this catchment (a sort of graphical conceptual model of catchment response). I think this will make the paper of more impact.

>>We have made the following conceptual figure and integrated it into the manuscript.

[Figure]

"Figure 1. Conceptual figure illustrating the four streamflow generation periods: Baseflow (A), Early melt (B), Melt (C), and seasonal Recession (D). Red arrows represent water depleted in heavier isotopes (for example, snow melt), and blue arrows represents water enriched in heavier isotopes (i.e. rain). Magnitude of flux (size of arrows), location of flux, distribution of saturated area (blue), and snow cover (white) vary between the four seasons. From left to right, top to bottom: Baseflow shows a minimal contribution of snow melt to subsurface and surface even during winter; during Early melt the snow recedes at the lower elevations first (red dashed line) and contributes to streamflow; the Melt period is characterized by retreating snow cover, fluxes dominated by rain input, and small snow contributions at high elevation; Finally seasonal Recession consists only of rain inputs. Asymmetric hillslopes with differing reservoir sizes and hydraulic conductivities may produce different responses during the different phases."

4) Finally, the language is overall good but sometimes there are not accurate terms, vague descriptions, and awkward sentences. Since there is at least one native speaker (I assume) in the authorship and other well-known and highly experienced researchers, I believe that the language issues could be easily fixed.

>> We have worked through the whole manuscript with this in mind.

5) At line 248 a 7-day moving average is mentioned and applied. In my experience, moving averages of such a long period produce i) a slight temporal shift of streamflow peaks; ii) a reduction of streamflow peaks. Please, comment this issue and its possible effect on the results, or adopt another smoothing method.

>> This is a good point. In this case we are just using the moving average to have a better image of what is going on and help determine the stream flow periods. i) There is not a temporal shift since the seven-day window extends 3.5 days forward and 3.5 days backward. Please see the Figure S9. ii) There is a reduction of the peaks, which helps us to interpret the processes. This information is used alongside the actual data not instead of it. We have made our method and purpose clearer in the text. The 7-day moving average reduces the peaks, highlighting the longer-term flow processes. We also use a baseflow indicator as the lowest of the 7-day moving average in a 10-day window based on the window determined in Mächler et al. 2021. These metrics are only used to classify the streamflow into the four characteristic periods.

6) At line 249 the Authors introduce the three streamflow periods B, E and M. Although I overall agree with them, I'm afraid that this is an arbitrary selection not based on an objective criterion (e.g., the use of a streamflow percentile threshold). Please, comment this issue and its possible effect on the results, or adopt another more objective selection method.

[Figure]

Figure S9 Illustration regarding the effect of a seven-day moving average on discharge measurements.

>> We have compared our periods and come up with some more quantitative guides. A condensed form of this table and points has been added to the manuscript.

"Table S1. Table showing characteristic flow conditions for each period, corresponding start and end dates. $Q_m$ is the median of raw data over the period. $Q_{m7}$ is the median of daily values of average 7-day moving average data. $Q_{r7}$ is the range of 7 day moving average data possible in one day. HH:MM is the median time of peak streamflow in a day. $Q_b$ is the median baseflow, calculated as the lowest of the 7-day moving average streamflow in a ten-day window (based on method in Mächler et al., 2021). $Q_f$ is the median of daily streamflow percentiles. $\Delta Q_{m7}$ is the change in streamflow according to a 7-day window around 1 day."

| Period | Start Date | End Date | $Q_{m1}$ [m³/s] | $Q_{m7}$ [m³/s] | $Q_{r7}$ [m³/s] | HH:MM | $Q_b$ [m³/s] | $Q_f$ [%] | $\Delta Q_{m7}$ [m³/s/d] |
|---|---|---|---|---|---|---|---|---|---|
| B0 | 01.Jan.16 | 31.Mar.16 | 0.59 | 0.75 | 0.02 | 09:08 | 0.64 | 15 | 0.01 |
| E0 | 31.Mar.16 | 06.May.16 | 2.44 | 2.56 | 0.11 | 22:47 | 2.20 | 58 | 0.10 |
| M0 | 06.May.16 | 24.Jun.16 | 9.72 | 10.08 | 0.37 | 15:43 | 8.94 | 93 | 0.11 |
| R1 | 24.Jun.16 | 27.Sep.16 | 3.87 | 4.62 | 0.15 | 06:57 | 3.57 | 74 | -0.12 |
| B1 | 27.Sep.16 | 18.Mar.17 | 1.05 | 1.01 | 0.02 | 20:59 | 0.89 | 27 | -0.01 |
| E1 | 18.Mar.17 | 06.May.17 | 2.23 | 2.23 | 0.06 | 00:07 | 2.10 | 52 | 0.02 |
| M1 | 06.May.17 | 02.Jun.17 | 6.84 | 7.26 | 0.33 | 18:20 | 5.91 | 84 | 0.31 |
| R2 | 02.Jun.17 | 27.Sep.17 | 4.04 | 4.22 | 0.15 | 13:45 | 3.10 | 70 | -0.08 |
| B2 | 27.Sep.17 | 08.Apr.18 | 1.01 | 0.99 | 0.02 | 04:06 | 0.94 | 26 | -0.01 |
| M2 | 08.Apr.18 | 31.May.18 | 7.85 | 8.32 | 0.31 | 12:28 | 7.55 | 87 | 0.19 |
| R3 | 31.May.18 | 01.Jan.19 | 4.59 | 4.64 | 0.09 | 13:20 | 4.47 | 74 | -0.06 |

The following quantitative rules were used to identify the periods, this is now included in appropriate form in the text
- The transition from M to R is always the peak of the 7-day moving average.
- Flow is increasing in E and M and decreasing in R and B.
- E is the first period of runoff generation after the winter recession, where the flow starts showing subdaily fluctuations, but no significant diel peaks are visible
- During M, there is a much higher diel variation in flow.
- Similarly, from R to B, the diel variation drops off.
- B has median of daily flow quantiles < 30 %, E around 50, M > 80 %, and R around 70 % or 75 %
- The change in flow of B per day (using 7-day average data to override event effects) is near 0 m³/s/d, for E around 0.1 m³/s/d, for M is positive and greater than that of E (0.1) up to 0.3 m³/s/d and for R is negative, down to near -0.1 m³/s/d.
- The median diel peak time of discharge is not as consistent, but it is late at night for E and closer to mid-day or late afternoon for M. During R, it is early afternoon or morning. For B, it does not exist (no diel peak).
- The range of diel values is low, around 0.02 m³/s for B, 0.1 m³/s for E or R, and high 0.3 m³/s for M.

7) 523-551. I suggest combining subsections 5.2, 5.3 and 5.4. I also suggest combining 5.6.3 with 5.1.

>> Thank you for this suggestion, it helped to restructure the discussion in point 3A.

8) Why can it be a calibration issue? What is this statement based on? Please, explain.

\>\> Their divergence is unexpected. However, at the end of base flow, it is very likely that there was not adequate water in the piezometers for accurate measurement. We were referring to this phenomenon. We have changed our wording in the manuscript.

**Reviewer 2**

Please find below a point-by-point response. In addition to general clarifications, we have 1) proposed aconceptual figure; 2) made the specific improvements to figures that you recommend; and 3) clarified the technical details around lapse rates and spring locations and counts. We have revised the manuscript and incorporated your minor comments and technical corrections.

**General comment:**

The paper makes an important contribution to increasing the understanding on flow dynamics in alpine catchments. This is done by combining different datasets, including stable water isotope tracers, EC and Temperature. Sampling of stable water isotopes in different storage compartments in the catchment (including vegetation) helps disentangle different hydrological processes (i.e., how does the catchment function). The figures are well prepared and the fact that the data is available will surely aid further research in alpine areas. The methods are clear and sound. Having said that and despite the interesting topic and relevance of the study I have concerns with the presentation and structure of the manuscript.   In general

**Detailed comments**

The Methods section would benefit from being more brief.

>> We condensed it by relying more on the referenced papers.

The results section should be revised and the sentences which discuss results moved to the discussion section.

>> Thank you for identifying some organizational problems. We have rearranged the text at several instances.

Interpretation of results from the multi-tracer approach would benefit from the addition of a conceptual diagram (see specific comments in E).

>> We include such a conceptual diagram (also benefitting from the suggestion of the first reviewer).

More references to international literature are needed in discussion and careful selection of figures to be referenced.

>> We have developed our connections to literature, as also suggested by the first reviewer.

Conclusions section needs strengthening.

>> We have done this in part by including a concluding paragraph with our lessons learned from our simultaneous examination of 4 tracers that is currently in the 5.6 section of the discussion.

Below I leave section by section comments which include both specific and technical comments.

**Detailed comments on Introduction:**

Line 51 - 70: It seems that you repeat in a way the study aim (see technical comments below). I suggest restructuring that part starting with a clear overarching aim and then

>> We've rephrased this section as follows:

> "To help fill some of the important knowledge gaps on elevational and seasonal drivers, this work uses high frequency tracer sampling to quantify drivers of the hydrologic response of a high elevation catchment throughout all streamflow periods, ranging from winter low flow, to different stages of the melt season and the autumn recession by compiling and complementing observational data from the intensively studied Vallon de Nant catchment in the Swiss Alps (Benoit et al., 2018; Giaccone et al., 2019; Ceperley et al., 2020;

Mächler et al., 2021; Michelon et al., 2021b; Thornton et al., 2021a; Beria et al., 2020b; Antoniazza et al., 2022).

The overall objective of this work is to examine dominant hydrological processes and associated flow paths in Vallon de Nant during different periods of the year through the lenses of four tracers: soil temperature, water temperature, electrical conductivity, and stable isotopes of water. The first tracer used in this study is the stable isotope composition of water, a natural tracer that has been extensively used to characterize snow hydrological processes (e.g. Beria et al., 2018), and is particularly useful to examine the interplay between different water compartments (rainfall, snowpack, springs, groundwater), recharge and evaporation processes (e.g. Sprenger et al., 2016). Electrical conductivity measurements as an additional tracer provides information on subsurface flow paths and water residence times in the subsurface (Cano-Paoli et al., 2019). Water temperature measurements can be used to quantify connectivity between water sources and the atmosphere (Constantz, 2008). And lastly, soil temperature measurements are used to identify periods of thermal insulation from the seasonal snow cover (Trask et al., 2020).

Using these tracers, we explore the origin of winter streamflow (the alteration and mixing of subsurface storage and from localized snow melt) (Floriancic et al., 2018; Hayashi, 2020), the dominant runoff processes that drive streamflow generation during early compared to late snow melt phase (Brauchli et al., 2017) and during the seasonal recession, the streamflow generated by shallow groundwater in the hillslopes and of alluvial or talus groundwater systems throughout the year (Hayashi, 2020). These explorations provide key transferable insights into the value of these four tracers for hydrologic process investigation that are relevant for comparable catchments."

**Detailed comments on Section 2**

(Section 2.1.) Study area: this section is unnecessary long. Is all the provided information then related to the study results? If not, you can reference previous papers for details on e.g., geology, and save space here.

>> We have carefully revised the section to condense it.

Lines 98-102: as written currently this paragraph reads like a discussion and it is not (e.g., "this topographic particularity might seem enough"). Please rewrite or move to discussion (e.g., to section 4.3.2.)

>> We have rewritten this as follows:

"The location of springs correlates with low slopes (see Figure S6 in Supplementary material), a topographic particularity explaining the location of springs along the right bank of the mainstream and within the grassy slopes in the west area of the catchment, where the slopes are low. In the same way, the absence of tributaries over the north-western parts of the catchment are related to steep slopes, explained by the large hydraulic conductivity and locally well-developed soils."

Figure 1: There is no "A" on the Figure 1, as indicated in the caption, please add. In caption "where the spring is picked up" – what do you mean by this? Also, can you make the legend of the B map, at present it is difficult to read.

>>Corrected. The note in the figure legend has been rephrased as: "Note that the AUBG spring location shows where the water is sourced from, even though it is sampled from a pipe at the Auberge weather station point, 800 m further north." and has been moved to the main text. The two legends have been placed together to make a single more comprehensive legend.

(Section 2.2.): Line 130 – 135: This seems like results (i.e., you analyzed the data to derive the annual average streamflow, for example). I would somehow include it in Results section.

>> section 2.2 (Meteorological and hydrological characterization of the study period) has been split into methods and a similar section starting the results section (3.1).

Line 139 – 143 refers to the meteorology. Given the title of section 2.1. starts with meteorology, I suggest you present this first to follow a logical order.

>> This is a good point, we have reorganized it as you suggest.

Detailed comments on Section 3, Methods

This section is very long. While the methods described are worth mentioning the descriptions are too wordy and make the reading monotonous. Here a few examples/suggestions on how to change that along with some minor technical notes.

>> We have rearranged this section to first be instrumentation (meteorological then hydrological), then tracers with subsections of 1) stable H- and O-isotopes of water, 2. water temperature, 3. conductivity, and 4. soil temperature, followed by additional data.

Line 153: Change "Water" (first word) with "Streamflow". Water is too generic; you sample water from various places.

>> Thank you for the reminder to be precise. We prefer to use a parenthesis for specificity as not all the water that we sample is streamflow: (from streams, springs, and piezometers)

Line 167: "The same borosilicate glass vials were also used for…" – this sentence is long and redundant. Define the type of vials and their volume once at the beginning of the subsection 3.1.1. and then only refer to them as "vials". It will save a lot of reading time.

>> done

Line 179: remove "sampled", as it is redundant

>> done

Line 219: I can count 5 springs on Fig.1, but T was measured in four. Can you spell out in which one you did not measure temperature?

>> We have clarified that we didn't measure temperature in the auberge spring (AUBG). We did not have access to the spring directly, only to a fountain where it was delivered by pipe. The information has been moved from the figure legend to the corresponding method section.

Subsection 3.1.2. Is extremely long and too much detail is given. These are standard methods, just give references and keep it tight. Only if you did something different or unique in the calculations then spell out which part that is to guide the reader.

>> Unnecessary details have been moved to a supplementary file.

Line 225: Reference "Figure 1" in "(At Auberge station).

>> Thank you for catching this omission

Line 227: see previous comment on excessive description of the vials

>> changed.

236: change "particularly useful for us" to "particularly useful for this study"

>> changed

244: you mention that the gridded data was useful for gap filling, but you do not say when did you have do gap fill and what % of your data that is. Please spell out.

>> We have removed the reference to gap filling. We did not use gridded data for gap filling. We did not need to gap fill for the purposes of this study and analysis. In an original justification of our data, we were just pointing out that our data were complementary to the gridded data set.

**Detailed comments on Section 4, Results**

The results section contains parts of discussion which should be moved to the discussion section (some examples below).The interpretation of results and to connect findings via different methods I suggest that you include a new figure, a conceptual model/diagram, which can also serve as a graphical abstract. This figure can be composed of 4 panels (A to D) and each one of them can describe graphically what have you learned with the multi-tracer method in A) The baseflow period, early melt period, melt period and seasonal recession period (as described in lines 248-249). This will aid the reader and will align with the aim "to provide transferable insights into the value of observed variables for hydrological process investigations in comparable catchments" (see lines 68-70).

>> Thank you for this suggestion. We've included a conceptual figure.

Lines 257-259: This is discussion. Move it there.

>> done

Figure 3: Add to the legend what B, E, M R mean. This saves the reader having to go back to the text and search for definitions. What is the faint blue line in the bottom panel? It looks as faint as the streamflow. At present this bottom panel is a bit confusing. Refer to Figure 1 for the abbreviations of e.g., soils, piezometer, spring IDs.

>>B, E, M, R have been added in the figure caption. The faint blue line in the bottom panel is the temperature at the outlet (HyS1), we tried to pick a better color to be more distinctive from the streamflow, and another color for streamflow to be more distinctive from PZ1 and PZ3. We propose to add a title and subtitles to the legend on the figure to help the reader to understand.

Line 317: sentence is too long. Change to "Sampled spring and ground water sources show varying correlations…" and add…"; having PZ1 the strongest correlation…"

>> done

Lines 334 – 344: example paragraph where the conceptual diagram will help the reader understand the description.

>> indeed, good idea.

Lines 353 – 359: this seems to belong to methods.

>> done

Lines 364- 368: Move this to the beginning of subsection 4.3.3. as this presents a more general finding. Then discuss the rest but please rewrite and remove the details that may belong to the methods.

>> good idea, done

(Section 4.1.)

374 – 379: This is more of a discussion, not a result.

>> We moved it to the discussion.

379-381: Sentence is too long. Rewrite or split in two.

>> According to your previous suggestion, part of this sentence have been moved to the discussion, thus we have broken it into two. In the results, only a sentence staying that we observe an event-scale lag in both streamflow and EC.

Figure 4: You do not make use of the subplot annotations A,B,C, etc. in the caption. Please edit your caption and organize by subplots. Also, when you reference this figure in the text you start with Figure 4F (Line 370). Please organize your subplots so they match the story line (e.g., if you first talk about EC then present the EC plot as Figure 4A). Same comment on the order applies to subsection 4.5.1.

>>We have added the annotations, A, B, C, and rearrange the text and subplots to correspond to each other.

Line 406: this is the first time you use lapse rate. Please define it very briefly earlier (e.g., in methods).

>> We have added a sentence regarding how we calculated lapse rate in the methods.

Subsection 4.5.1. is structured poorly. You mention a lot of Figures but make little use of them.

>> We have developed and organize the section further.

Figure 5- 7: Similarly to Figure 4 you make subplot annotations A,B,C,..but do not use them in the caption. Also, is there a way to skip one or two of the subplots (and move them to supplementary materials) and consequently make the plots larger and horizontally oriented. At present it is very difficult to read them if you don't print the manuscript.

>>We have added the annotations, A, B, C, and rearrange the text and subplots to correspond to each other. We have removed some subfigures so that they can be horizontal. This is a good suggestion.

Subsection 4.5.2. Line 427-428: if the variations are similar between the different stable water isotopes and you will comment only on delta 18 O then why don't you save space and present those in supplementary material?

>> done.

Line 439: remove "it's"

>> done.

Subsection 4.5.3., 4.5.4.

Make more reference to your Figures in the text. And similarly, to Subsection 4.5.2. – move to supplementary materials the figures you do not reference.

>> Done We have.

Subsection 4.5.5.: Line 480: The first sentence is not a good start of a subsection. Remove is and simply reference the figure in brackets in a rewritten first sentence. In the first sentence present the general idea/finding from lc-excess. Second sentence at present is too difficult to follow as it is too long– split in two.

>> Done. They now read as:

"The range of values of LC-excess for the rainfall samples are related to the spread around the evaporation line (Figure 4F). We see that the median value of the snowpack samples is close to the reference for rainfall (0 ‰). Our observations validate Beria et al. (2020)'s review of snowpack data for entire snow seasons and does often not show a significant deviation from median values from the reference precipitation value."

You discuss the results in terms of interpretation of the data but do not make a reference to other literature. In this regard the discussion needs more work and strengthening.

>> We developed our discussion in terms of comparison and discussion of international literature.

Also, the terminology used in this section sounds a bit awkward. Examples below:

>> We have carefully edited and revised all the text with attention to this critique.

Line 515: "enrichment in light isotopes during winter" – which are the light isotopes of hydrogen and oxygen? Do you mean "stable water isotope signal becomes more depleted during winter"?

>> We've revised it to read:

> "However, we measured diverging isotopic ratios in two springs, one demonstrating an enrichment in the heavy H- and O-isotope composition (AUBG) and the other a depletion in these isotopes (BRDG) during winter (Figure 5)."

Lines 516, 534: "light isotopes" – same comment. Revise the terminology

>> We've revised it to read:

> "Winter melt processes contributing to the groundwater system throughout the winter would lead to such a depletion in the heavy H- and O-isotopes. "

Section 5.2. would benefit from reference to the conceptual diagram I suggest above.

>> Done

Lines 555 – 558: this sentence is too long and difficult to follow. Rewrite.

>> We've rewritten it as:

> "During M2, both BRDG and PZ3 temperature is correlated with streamflow variations although one is a positive anomaly (BRDG) and the other negative (PZ3).  The positive anomaly measured at BRDG suggests a snowmelt input that is heated up before infiltration, which is understandable because BRDG collects snowmelt from the nearby riparian area and steep slopes facing west in direct sun exposure. In contrast, the negative anomaly at PZ3 suggests the melted snow is directly infiltrating as it begins with the melt period and ends when the area is free of snow (200 m from soil temperature sensor at 1,530 masl), suggesting that the infiltration of snowmelt is local. "

Line 578: "streamflow isotopes" would sound better as "isotope signal in streamflow"

>> We've change this to "isotope composition of the water in the streamflow". Furthermore we have corrected the language imprecision or colloquialism throughout the manuscript.  We have used " observe" for what we see and "measure" for what we measure in our revision.

Line 611: "which locations become more enriched in heavy isotopes" – the location does not become more enriched, it is rather the water in those locations shows a more enriched isotope signal. Please rewrite.

>> We've rewritten this to read:

> "However, this requires year-round time series to measure where and when water becomes enriched or depleted. "

The conclusions section needs strengthening. Mention what are the implications of your findings and how is this study important in terms of better understanding the water dynamics of Alpine catchments.

>> Done

Lines 650 – 655: This is not a strong ending of a conclusion section, and it does not reflect the efforts employed in developing and executing the field study. I suggest a small subsection is dedicated on delta 17 O somewhere earlier in the manuscript and all the findings are concentrated there.

>> Thanks. This has been moved to a section of the discussion following "the added value of EC": "The added value of d$^{17}$O and $^{17}$O-excess".

**Reviewer 3**

We are pursuing deeper analyses like those that you recommend for subsequent publication. This paper is really meant to focus on the added value of this set of observations directly from the field. We have developed our discussion and conclusion to address the imbalance between the sections that you describe. We also have refined our conclusion to be concrete rather than a vague claim that you perceived of "more must be done".

**General comments**

I am reviewing MS HESS-2022-48 by Michelon et al. on "Studying the dynamic of a high alpine catchment based on multiple natural tracers". The authors report on a quite comprehensive (3 years long) dataset of temperatures, water electrical conductivity (EC) and stable isotopic compositions in a range of (eco)hydrological compartments (stream, springs and vegetation) of the Swiss "Vallon de Nant" high-altitude headwater catchment for the investigation of "dominant hydrological processes".

The MS is well written, easy to follow and of appropriate size, and the figures are well crafted (despite incomplete legends). Finally, the MS obviously fits the scope of HESS well.

>> Thanks for this overall positive assessment.

The main outcome of the study seems to be that "more must be done", possibly using the same types of observations (using a multi-tracer framework by coupling EC, temperature, and isotopic analyses) but at higher temporal resolution to achieve the goal. There is a general imbalance between the amount and quality of data collected and the quite superficial analysis presented. For a deeper analysis, the authors could, for instance test a simple two-end member approach as per the seasonal origin index, with rain vs snow as endmembers to investigate tree water uptake seasonal use?? Also, some of the co-authors have extensive knowledge of catchment hydrology process-based modeling; the data may also call for such an application to calculate e.g., transit time distribution etc.

>> Thank you for this feedback and suggestions. See also our comment on the start of our response to reviewer 3.

**Detailed comments**

TITLE: The dynamic in what?

>> This is a good point. Alpine catchments are considered as being particularly dynamic systems (see the introduction of the Mächler 2021 paper about the same catchment for a more complete explanation) and we would like to emphasize that in the title. With the support and recommendation of the editor, we changed the title to:

"Hydrodynamics of a high Alpine catchment characterized by four natural tracers."

ABSTRACT: L16. This goes for pretty much all environments displaying dynamics in water δ, right?

>> This is true but is especially apparent and relevant in this environment with clear seasonal cycles influencing these tracers. In any case, the abstract has been completely reworked according to suggestions of other reviewers and this sentence no longer appears. In the one location still using the phrase "such environments", we have added "dynamic" to clarify the characteristic of the environment that necessitates year-round water sampling.

INTRODUCTION: L55. I would not refer to unpublished works, also because you have quite a few papers to cite from, some of which are only 1 year old.

>> the reference list has been updated upon resubmission and any still unpublished works have been removed.

L64. It is your "overall" objective, not (one of) your specific one(s).

>> indeed, we have changed the wording.

L65-66. Or a mix of the two? Or is it implied here?

>> we have clarified this by changing the wording to: "the alteration and mixing of subsurface storage and from localized snow melting"

METHOD: L186. It comes as a surprise at this point of the MS that you would sample from the vegetation. What is the purpose? This should be introduced somewhere earlier.

>> we have removed this and saved it for a subsequent analysis (and manuscript) focused on hydrological interactions with vegetation.

L190. There is just one delta notation, used for different elements and their stable isotopes. Please rephrase.

>> done

L196-197. "and data from the last 6 injections were kept"

>> modified

L209-210. To be "less temperature sensitive" and "convey additional information on evaporation processes and on climatic conditions" seems to be contradicting… please rephrase/elaborate

>> We have reworded this:

> "Both d-excess and $^{17}$O-excess are known to respond to relative humidity during evaporative processes, but $^{17}$O-excess may be less temperature sensitive (Surma et al., 2021; Bershaw et al., 2020) than d-excess and thus changes in its composition may be more sensitive to net evaporation, including secondary evaporation, as well as the meteorological conditions even when they would be invisible with d-excess (Risi et al., 2010)."

L219. Why "GRAS" and "ROCK" are in capital letters?

>> We are using four letter capital abbreviations for names of sampling locations. We have added a sentence in the site description.

L241. "long"?

>> We have replaced with the length of series.

L242. "Influenced by the low"

>> We have replaced "influenced" with "limited"

Figure 3. It is a nice picture, but could use more info: e.g., name each of the three different panels. It is difficult to understand what "bottom" refers to (i.e., is it the bottom part of the top panel, or the actual bottom graphics?). The caption should be as self-explanatory as possible, therefore define also here what "B", "E", "M", and "R" mean. Bottom graphics: it is difficult to differentiate between streamflow data and the water temperatures in gray colors. Maybe move the 2nd y-axis legend ("Streamflow [mm/day]") a bit down so that it faces lower values, e.g. [0-20 mm/day] in each panel?

>> Thank you for your feedback. We have incorporate many of your suggestions and reworked the figure.

RESULTS: Usually, (campaign) results are related in past tense to differentiate with literature findings and general statements (made in present tense). Also avoid using "shows this and that…". Use a more direct formulation, e.g., in L251: "The baseflow period extends from the end of September to early spring (mid-March to beginning of April) and shows a streamflow of around 1 mm/d only" vs. "The baseflow period extends from the end of September to early spring (mid-March to beginning of April) with streamflow values of approx. 1 mm/d only…"

>> Thank you for this feedback. We have edited and restructured the results section.

L257-259. Could be a nice discussion and moved there.

>> done

L260. "due to an important water input from snowmelt." Do we need this piece of info again?

>> deleted

L260-262. But isn't it because there was no early melt period that the melt period started sooner in 2018?

>> Perhaps, we have reworded this to make this clear.

L317-319. Avoid such formulation and just start with the actual results, e.g., "Correlation between spring and air temperature at the Auberge station was source-specific…".

>> Indeed, this sentence has been modified as to recommendations from a previous reviewer.

Also: BRDG and ICEC acronyms are not defined

>> Thank you for pointing this out, we now define all point names in the site description section.

L320. I am not a specialist, but does a spring have a "volume", strictly speaking?

>> You are right, we have clarified this to refer to the magnitude of flow out of the spring.

L325. "temperature [curve]"

>> We have changed this to: "The shape of the curve of temperature fluctuations of the …"

Table 1. Why it the maximum stream temperature not reported?

>> This is an oversight on our part, we have added it.

L353-354. The Lag "L" should have its own equation reported in section 3.2. It is difficult, at least to me, to understand what was done here…

>> As noted, we reported this in Appendix 2.

L401-402. Such an intro within the result section is not needed. Instead give the results and point to the figure/table for substantiation.

>> We rigorously edited the results sections so that it is more to the point.

L403. Please define "lapse rate".

>> done.

Fig. 5 vs Fig 6 & 7. Why are you connecting the dots for springs, therefore implying linear interpolation, when you do not do this for e.g., streamflow or rainfall?

>> We felt that this depiction made it easier to distinguish and compare the fluctuations at the different springs. We have reconsidered the lines and are clearer about their purpose in the text.

Also replace "δD" by "δ2H" throughout, please. I would remove all unnecessary material, that is all variables that are not described in the text. This would also make the figures more reader friendly.

>> We considered moving some material from the figures to the supplementary files but decided it was more valuable together. We have however removed some material from the figures and made them more visually appealing.

L427-428. What is shown in Fig. 5 should be discussed, so why showing both δ2H and δ18O when only δ18O is discussed?

>> Indeed, we have considered moving them to the supplementary material but decided that we do touch on them in the text and it is more illustrative to keep the figure complete. We have however reformatted the figure to make it more visually appealing.

L431. "with [lower] isotopic values"

>> thank you

L455. "and a significant decrease in the [isotopic composition]".

>> thank you

What about the rest of the variables this time (δ18O & δ17O)?

>> indeed, we said we focus on δ18O, we have clarified this.

L475. "local scale process information". Name some examples.

>> Here we are referring to the interplay of mixing verses fractionation in the immediate vicinity of sampling. In order to interpret these results, we must consider the storage and release processes on a catchment scale.

L481. What evaporation line? Here you are looking at the deviation from the LMWL...

>> true, this is a typo on our part

L484-485. I do not understand. Please rephrase. Do you mean to say that larch trees xylem water has low Lc-Excess values, meaning they sample from water departing isotopically from meteoritic water sources (i.e., evaporatively enriched soil water)?

>> You are correct, this is an oversimplification on our part. We have removed discussion of vegetation for the purposes of this paper and focus on it in a subsequent paper.

L486. "negative median [lc-excess] value"

>> Thank you

L492-496. 17-Excess measurements are very tricky, and I ask myself if differences to other studies have to do with the analysis technique used, i.e., mass vs. laser spectrometer?

>> We do not believe that this is the case, as for both mass and laser spectrometers, the calibrations are done with standards that are calibrated with the MS-analytical approach. However, memory effects can influence the 17-O measurements. We have added more discussion regarding the risk of this in the respective methods and discussion sections as appropriate.

L508-509. This is not needed.

>> Indeed, we reworded this to be a topic sentence.

DISCUSSION: L514-515. This I never read 🙂: please change to e.g., "an enrichment (depletion) in heavier stable isotopes at AUBG (BRDG)…"

"Such a depletion by heavier isotopes"

>> We've revised it to read:

> "However, we measured diverging isotopic ratios in two springs, one demonstrating an enrichment (AUBG) and the other a depletion (BRDG) in the heavy H- and O-isotopes during winter (Figure 5)."

L534-535. "Although the δ2H, δ17O and δ18O annual medians of AUBG, ROCK, BRDG and ICEC decrease with elevation"

>> Thank you

L622. Again, I doubt that the δ17O and 17O-Excess add value to the already measured δ2H and δ18O time series…

>> This discovery would already be an added value as not that many δ17O and 17O-Excess have been made so far. We emphasized this.

CONCLUSION: L650. The reader still does not know what you mean by "local-scale snow hydrological processes"

 >> We moved this conclusion into a discussion section and elaborate more extensively and explain what we are referring to here.

---

## Author Response (AR2)

Dear Editor, Dear Reviewers

We would again like to thank you for handling our paper and giving additional advice. Below, we respond to the latest round of requests and mention the changes we've made or in some cases a justification for the lack of change. Our responses are introduced by double carrots >> after each reviewer comment.

Thanks to the reviewers' comments, the quality of the manuscript has further improved, and we believe that it is now ready for publication.

On behalf of all authors,

Natalie Ceperley

**Reviewer 1**

We are very thankful for the time and feedback you gave us. Please find below a point-by-point response to your specific comments.

Language. There are still typos, tense inconsistencies (switches from present to past in the same paragraph or even in the same sentence), some convoluted sentences. Please, have the language carefully revised by a native speaker or by a proficient speaker.
>> We have made some changes to awkard sections, in particular to the abstract and methods section.

2. The new title is not so convincing to me. It seems that the catchment is characterized by four tracers, and not the hydrodynamics. I suggest looking for a more attractive and informative title. For instance, I would mention something like "Interplay of hydrological processes in a high elevation Alpine catchment based on four tracers…"…or "Multiple tracers allow understanding of the interplay of hydrological processes etc."
>> After discussion between co-authors, we have decided to retain the focus on hydrodynamics and not further modify the title and keep it as:

*Hydrodynamics of a high Alpine catchment characterized by four natural tracers.*

76. I suggest including here, or later, the specific research questions that the work addresses.
>> We think the last two paragraphs of the introduction do this already. Repeating the objective in the form of a question seems redundant.

179 and 361. Sorry, the use of the moving average is still not clear to me, and I am still afraid of the possible bias it could lead to. Please, clarify this point.
>> We've added more explanation that was in our reply to reviewers previously following the last round of revisions but not elaborated on in the paper:

*"To guide the analysis of the streamflow response throughout the year, we analyzed the different streamflow periods in detail (Section 3.1) based on the 7-day moving average streamflow data (Qm7) and the daily change of Qm7, called ΔQm7. This moving average does not introduce a temporal shift since the seven-day window extends 3.5 days forward and 3.5 days backward, however it does reduce the peak*

*flow. Its interpretation alongside the high-resolution streamflow data does not introduce a bias but rather highlights the slower processes."*

952-953. Please, explain better (and perhaps include references) your point about sampling snow vs. snowmelt.
>>By rearranging the paragraph and combining it with the 4.4.4 O17 paragraph and adding more references, we believe this point is now clear:

*"While so far not reported for measurements of $\delta^{17}O$, it is known that under certain environmental conditions, the snowpack will experience melting, evaporation, and refreezing, forming a firn rather than snow. This may influence the $^{17}O$-excess measured for either fresh snow or the final snowpack (see, for example, Risi et al., 2010). Although, much laboratory time was devoted here to the measurement of $\delta^{17}O$ and $^{17}O$-excess, we gained few insights or specific added value as we hoped for documenting the influence of local-scale snow dynamics, specifically the variation in space and time of accumulation, transport, storage, melt and sublimation, on hydrological processes, except some unvalidated potential to distinguish glacier melt from snowmelt when combined with temperature measurements. Perhaps our measurements would have been more relevant if fresh snow had been sampled instead of the snowpack or if more relevant reference data were available."*

Subsections 4.4.3 and 4.4.4 are very short. I wonder if they could be merged or included in other subsections.
>> We merged the O17 paragraph above to the other isotope paragraph.

Conclusions: I'd avoid the over-use of the term "reset".
>> Thank you, we've reworded or elaborated on all the mentions of "reset" in the conclusion.

Minor comments and technical corrections
17. Store = storages?
>> We changed this in the abstract to:

*"Although diverse water sources and flow paths generate streamflow in the world's "water towers" emerge from these two driving inputs, a detailed process understanding remains poor."*

186. Although this is quite common, this is not the correct terminology. I suggest using stable isotopes "in" water, or "of oxygen and hydrogen".
>> We've modified this expression throughout the paper:

*"Stable isotope composition of water"*

217. "von", with no capital letter.
>> Thank you – we've made it lower case.

256. Remove "familiar".
>> Thank you .

318-319. Include references, please.

>> We've added more explanation and references:

> *"Although it is common to study the effect of snow cover on near surface soil temperature especially for prediction of microclimates and habitats (Rixen et al., 2008; Freppaz et al., 2018; Giaccone et al., 2019), the inverse focus is also relevant and soil temperature is a good proxy for snow cover (Bender et al., 2020; Staub et al., 2015), making distributed observations of soil temperature particularly useful for observing the expansion and contraction of snow cover."*

721. Comparison of what? Please, specify.

>> We've reworded this for more clarity:

> *"Comparison of similar tracer studies in Alpine regions"*

726. What is the headwater status? Please, clarify.

>> We've reworded this for more clarity:

> *"The slightly lower values can be explained by the proportion of snow to rain in the specific headwater region of this study relative to other regions."*

797. What do the Authors mean by "prevalent"? Please, explain.

>> We've reworded this for clarity:

> *"The early melt period is rarely discussed in the literature (for a model-based example, see He et al., 2015), despite it being its importance in terms of water supplied to the catchment and prevalence across Alpine regions, and the streamflow during this period remains challenging to model (see Figure 9 in Brauchli et al., 2017; or Figure 3 in Thornton et al., 2022)."*

851. Give reference for the old water paradox.

>> We've added references:

> *"Our measurements clearly suggest that this so-called "old water paradox" does not hold for some rainfall-generated streamflow responses during the recession period (Mcdonnell, 1990; Mcdonnell et al., 2010)."*

**Reviewer 2**

In my review of the initial submission, I raised my main concern that the study of Michelon and colleagues was a bit descriptive and could benefit from a modeling component. Partly in response to this - but I cannot be sure - the authors added a conceptual drawing about the succession of streamflow generation periods at their experimental site. This is a first step, however the contrast between the amount and quality of the data and the qualitative interpretation remains. The authors argue elsewhere that their isotope data (more precisely their dynamics) are too complex for a straightforward interpretation. I agree with this statement but in those situations a model application could certainly shed light.

Since my opinion does not seem to be shared by the other reviewers and that the data alone is worthy of publication (for others to model them), I will personally not object to its publication.

>> Thank you for your recommendation. While we agree that a modelling approach would help improve our quantitative interpretation of the phenomenon, we believe this is outside the scope of the current paper. We will add a sentence in the future outlook that emphasizes your recommendation:

> *"A future investigation in this location could render our conclusions more quantitative*
> *by using a modelling approach to explore our observations."*

**References referred to in this letter**

Bender, E., Lehning, M., and Fiddes, J.: Changes in Climatology, Snow Cover, and Ground Temperatures at High Alpine Locations, Front Earth Sc-Switz, 8, 100, 10.3389/feart.2020.00100, 2020.

Brauchli, T., Trujillo, E., Huwald, H., and Lehning, M.: Influence of Slope-Scale Snowmelt on Catchment Response Simulated With the Alpine3D Model, Water Resources Research, 53, 10723-10739, 10.1002/2017wr021278, 2017.

Freppaz, M., Pintaldi, E., Magnani, A., Viglietti, D., and Williams, M. W.: Topsoil and snow: a continuum system, Applied Soil Ecology, 123, 435-440, 10.1016/j.apsoil.2017.06.029, 2018.

Giaccone, E., Luoto, M., Vittoz, P., Guisan, A., Mariéthoz, G., and Lambiel, C.: Influence of microclimate and geomorphological factors on alpine vegetation in the Western Swiss Alps, Earth Surface Processes and Landforms, 44, 3093– 3107, 10.1002/esp.4715, 2019.

He, Z. H., Tian, F. Q., Gupta, H. V., Hu, H. C., and Hu, H. P.: Diagnostic calibration of a hydrological model in a mountain area by hydrograph partitioning, Hydrology and Earth System Sciences, 19, 1807-1826, 10.5194/hess-19-1807-2015, 2015.

McDonnell, J. J.: A Rationale for Old Water Discharge Through Macropores in a Steep, Humid Catchment, Water Resources Research, 26, 2821-2832, 10.1029/WR026i011p02821, 1990.

McDonnell, J. J., McGuire, K., Aggarwal, P., Beven, K. J., Biondi, D., Destouni, G., Dunn, S., James, A., Kirchner, J., Kraft, P., Lyon, S., Maloszewski, P., Newman, B., Pfister, L., Rinaldo, A., Rodhe, A., Sayama, T., Seibert, J., Solomon, K., Soulsby, C., Stewart, M., Tetzlaff, D., Tobin, C., Troch, P., Weiler, M., Western, A., Wörman, A., and Wrede, S.: How old is streamwater? Open questions in catchment transit time conceptualization, modelling and analysis, Hydrological Processes, 24, 1745-1754, 10.1002/hyp.7796, 2010.

Risi, C., Landais, A., Bony, S., Jouzel, J., Masson-Delmotte, V., and Vimeux, F.: Understanding the O-17 excess glacial-interglacial variations in Vostok precipitation, J Geophys Res-Atmos, 115, 10.1029/2008jd011535, 2010.

Rixen, C., Freppaz, M., Stoeckli, V., Huovinen, C., Huovinen, K., and Wipf, S.: Altered snow density and chemistry change soil nitrogen mineralization and plant growth, Arctic, Antarctic, and Alpine Research, 40, 568-575, 10.1657/1523-0430(07-044)[RIXEN]2.0.CO;2, 2008.

Staub, B., Marmy, A., Hauck, C., Hilbich, C., and Delaloye, R.: Ground temperature variations in a talus slope influenced by permafrost: a comparison of field observations and model simulations, Geographica Helvetica, 70, 45-62, 10.5194/gh-70-45-2015, 2015.

Thornton, J. M., Therrien, R., Mariéthoz, G., Linde, N., and Brunner, P.: Simulating fully-integrated hydrological dynamics in complex Alpine headwaters: potential and challenges, Water Resources Research, 58, e2020WR029390, 10.1029/2020WR029390, 2022.

---

## Author Response (AR3)

Dear Editor,

We would again like to thank you for handling our paper and giving additional advice. The previous two letters describe in detail all the changes that we made in the last two revisions (minor and major). The only additional change we made to produce the final manuscript was adding page numbers. Additionally, according to the manuscript guidelines, we removed all tables and figures from the main text. The figures are now only included in the separate zip file. The figure captions are sequential at the end of the manuscript. The tables and their captions are included after the figure captions. We have tested all the figures with a color-blind lens and do not believe that they pose major problems.

Please refer to the point-by-point revisions and the marked-up track changes from the rounds of minor and major revisions if necessary to follow the previous modification.

On behalf of all authors,

Natalie Ceperley